# Ufbp1 promotes plasma cell development and ER expansion by modulating distinct branches of UPR

Huabin Zhu[1], Brinda Bhatt[1], Sathish Sivaprakasam[1,7], Yafei Cai[2], Siyang Liu[1], Sai Karthik Kodeboyina[3], Nikhil Patel [4], Natasha M. Savage[4], Ashok Sharma[3], Randal J. Kaufman [5], Honglin Li[1,6] & Nagendra Singh [1,6]

The IRE1α/XBP1 branch of unfolded protein response (UPR) pathway has a critical function in endoplasmic reticulum (ER) expansion in plasma cells via unknown mechanisms; interestingly, another UPR branch, PERK, is suppressed during plasma cell development. Here we show that Ufbp1, a target and cofactor of the ufmylation pathway, promotes plasma cell development by suppressing the activation of PERK. By contrast, the IRE1α/XBP1 axis upregulates the expression of Ufbp1 and ufmylation pathway genes in plasma cells, while Ufbp1 deficiency impairs ER expansion in plasma cells and retards immunoglobulin production. Structure and function analysis suggests that lysine 267 of Ufbp1, the main lysine in Ufbp1 that undergoes ufmylation, is dispensable for the development of plasmablasts, but is required for immunoglobulin production and stimulation of ER expansion in IRE1α-deficient plasmablasts. Thus, Ufbp1 distinctly regulates different branches of UPR pathway to promote plasma cell development and function.

[1] Department of Biochemistry and Molecular Biology, Augusta University, Augusta, GA 30912, USA. [2] College of Animal Science and Technology, Nanjing Agricultural University, 1 Weigang, Nanjing 210095 Jiangsu Province, China. [3] Center for Biotechnology and Genomic Medicine, Augusta University, Augusta, GA 30912, USA. [4] Department of Pathology, Augusta University, Augusta, GA 30912, USA. [5] Degenerative Diseases Program, Sanford Burnham Prebys Medical Discovery Institute, La Jolla, CA 92307, USA. [6] Georgia Cancer Center, Augusta University, Augusta, GA 30912, USA. [7] Present address: Department of Cell Biology and Biochemistry, Texas Tech University Health Sciences Center, Lubbock, TX 79430, USA. Correspondence and requests for materials should be addressed to N.S. (email: nasingh@augusta.edu)

Following encounter with cognate antigen, naive B cells proliferate and differentiate into antibody-secreting cells (ASCs). Two types of ASCs develop during B cell responses: short-lived plasmablasts and long-lived plasma cells. Plasmablasts are generated early during the B cell response and produce low-affinity antibody against antigen[1]. B cells entering the germinal centers of secondary lymphoid follicles differentiate into plasma cells[2]. Plasma cells are post-mitotic cells, representing the end stage of the B cell differentiation program, and soon after their development home to the bone marrow and reside within specialized niches. High-affinity antibodies secreted by plasma cells play a critical role in the neutralization of pathogens. Therefore, understanding the molecular and cellular mechanisms regulating plasma cell differentiation and function is important in designing vaccines to generate better humoral responses and approaches to target harmful plasma cells.

Differentiation of B cells into plasma cells is regulated by the coordinated expression and repression of multiple transcription factors. The transcription factors Pax5, Bcl-6, and Bach2 are expressed in B cells, support the transcriptional program that maintains B cell identity, and suppress plasma cell differentiation[3–7]. On the other hand, the transcriptional programs induced by BLIMP1, IRF4, and XBP1 extinguish B cell genes and stimulate differentiation of plasma cells[8–18]. Other transcription factors such as IRF8 and PU.1 negatively regulate plasma cell differentiation by stimulating expression of Bcl-6 and Pax5[19]. Similarly, microphthalmia-associated transcription factor inhibits plasma cell development by suppressing IRF4 and BLIMP1[20]. In general, plasma cell-associated transcription factors oppose the function of the transcription factors responsible for maintaining B cell identity and vice versa.

Accumulation of unfolded proteins in the endoplasmic reticulum (ER) lumen results in ER stress. Cells respond to ER stress via activation of unfolded protein response (UPR) pathway. Three UPR pathways: inositol-requiring transmembrane kinase/endonuclease 1 (IRE1), PKR-like ER protein kinase (PERK), and activating transcription factor 6 (ATF6) sense the ER stress, induce signaling to upregulate expression of chaperones, and expand ER network leading to enhancement of protein folding capacity of ER. The expanded ER network facilitates proper folding and secretion of a large amount of secretory proteins. Thus, UPR pathway plays a central role in development and function of secretory cells. Plasma cells are secretory cells. Ligand-driven model suggests that during ER stress, interaction of ER luminal domains of IRE1α and PERK with misfolded proteins plays an important role in their activation[21,22]. Since ER luminal domains of PERK and IRE1α share similar conserved residue and mutational analysis suggest similar requirements for their activation, it is surprising that during development of plasma cells, IRE1α is robustly activated, whereas activation of PERK is suppressed[16,23–26]. The mechanism and significance of PERK suppression in developing plasma cells are not fully understood.

The endonuclease activity of IRE1α excises a 26-nucleotide segment from the XBP1 mRNA. The splicing shifts the reading frame, resulting in the translation of full-length XBP1, which translocates into the nucleus and transcribes genes involved in ER expansion, protein folding, protein synthesis, and transcription of secretory IgM in plasma cells[13,16,27–29]. In the absence of XBP1, plasma cells develop normally but due to defective expansion of ER network and *Igh* mRNA processing, show impaired ability to secrete immunoglobulins[8,25,30]. However, identity of XBP1 target/(s) that play a pivotal role in the expansion of ER in plasma cells remains poorly characterized.

Ubiquitin-fold modifier 1 (Ufm1) is a ubiquitin-like polypeptide that is post-translationally conjugated to target proteins via the ufmylation process and thereby modifies their function. Similar to ubiquitinylation, ufmylation is a three-step biochemical reaction catalyzed by specific E1 (Uba5), E2 (Ufc1), and E3 (Ufl1)[31–33]. Ufm1-binding protein (Ufbp1, DDGRK1, C20orf116, or Dashurin) is the first identified target of the Ufm1 pathway[33,34]. Anomalies in the ufmylation pathway are associated with neuronal diseases[35–39], spondyloepiphyseal dysplasias[40], developmental defects[41], and blood disorders[42,43]. We and others have recently published that Uba5, Ufl1, and Ufbp1 play a key role in the survival of hematopoietic stem cells and hematopoiesis[44–46]. Ufmylation pathway is upregulated following pharmacological ER stress and protects cells under these conditions[34,47]. Nonetheless, it is unknown whether Ufbp1 is a general regulator of all three branches of UPR pathway or plays different roles at different branches of UPR pathway.

In this study, we show that Ufbp1-mediated suppression of PERK has an essential function in the differentiation of naive B cells into plasma cells. In addition, Ufbp1 is upregulated downstream of IRE1α /XBP1 pathway to critically enforce the expansion of ER network and function of plasma cells.

## Results

**Ufbp1 regulates humoral immune responses**. To study the role of Ufbp1 in the biology of B cells and plasma cells, we crossed *Ufbp1^flox/flox* (*Ufbp1^F/F*) mice with *CD19^cre/+* (*CD19^cre*) mice and obtained *Ufbp1^F/F CD19^cre* mice. *Ufbp1^F/F CD19^cre* mice were born and remain healthy for rest of their life. B cells (B220+CD19+ cells) were present at comparable frequencies and numbers in the spleen and lymph nodes of *Ufbp1^F/F CD19^cre* and *Ufbp1^F/F* mice (Fig. 1a, Supplementary Fig. 1a–d). Expression levels of CD19 and B220 were also comparable between these two lines of mice. Frequencies of IgM^low IgD^high, IgM^high IgD^low, and IgM^high IgD^high cells were also similar between *Ufbp1^F/F CD19^cre* and *Ufbp1^F/F* mice (Supplementary Fig. 1e). Naive B cells from *Ufbp1^F/F* mice expressed low levels of Ufbp1 protein, which was upregulated ~10-fold following lipopolysaccharide (LPS) treatment (Supplementary Fig. 1f). In contrast, Ufbp1 protein expression in naive B cells and LPS-activated B cells from *Ufbp1^F/F CD19^cre* mice was ~5- and ~17-fold lower respectively than *Ufbp1^F/F* mice, indicating efficient deletion of *Ufbp1* in the B cells of the former (Supplementary Fig. 1f). Notwithstanding, with normal number of B cells, serum immunoglobulins of all isotypes were present at significantly lower levels in unimmunized *Ufbp1^F/F CD19^cre* mice than *Ufbp1^F/F* mice (Fig. 1b). To test the role of Ufbp1 expression in B cells in T-cell-dependent (TD) antibody responses, 4-hydroxy-3-nitrophenylacetyl (NP) conjugated to keyhole limpet hemocyanin (KLH; NP-KLH) precipitated on alum was injected intraperitoneally (i.p.) into *Ufbp1^F/F CD19^cre* and *Ufbp1^F/F* mice. At all the time points following immunization (days 7, 21, and 90) compared to *Ufbp1^F/F* mice, *Ufbp1^F/F CD19^cre* mice exhibited significantly lower serum levels of NP-specific antibodies of the isotypes, IgM, IgG1, and IgG3 (Fig. 1c). NP-specific antibodies of IgG2a and IgG2b were also present at significantly reduced level on day 21 and 90 in sera of *Ufbp1^F/F CD19^cre* mice than *Ufbp1^F/F* mice. In addition, following immunization with T-cell-independent (TI) antigen, NP-ficoll, *Ufbp1^F/F CD19^cre* mice also generated significantly reduced quantities of NP-specific antibody of the IgG3 isotype than *Ufbp1^F/F* mice (Fig. 1d). Taken together, these data show that Ufbp1 expression in B cells is necessary for the generation of TD as well as TI antibody responses.

**Defective generation of plasma cells in *Ufbp1^F/F CD19^cre* mice**. Due to the significant reduction in both serum immunoglobulins and antibody responses, the number of plasma cells was evaluated in *Ufbp1^F/F CD19^cre* mice. First, sorted splenic naive B cells (B220+CD138−) and bone marrow plasma cells (B220−CD138^high) from wild-type (WT) mice were compared for expression of

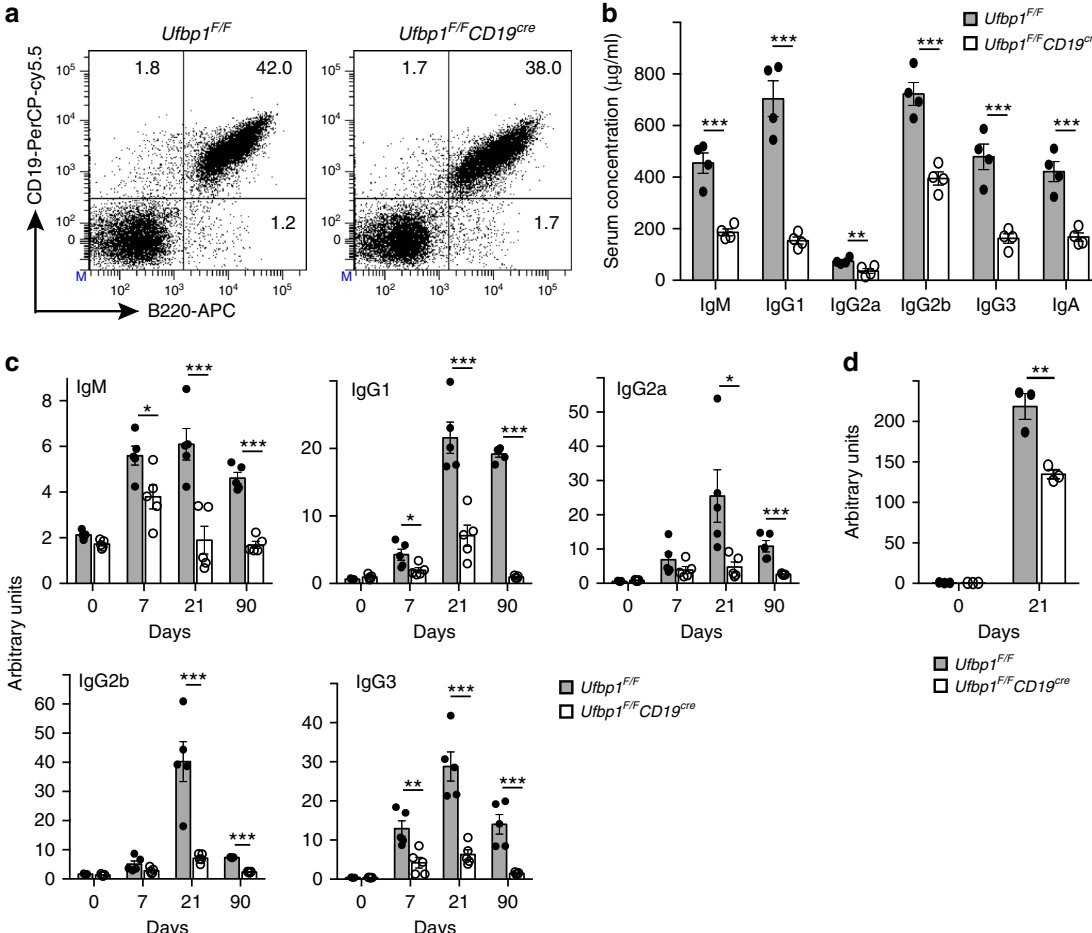

**Fig. 1** Defective humoral response in *Ufbp1^F/F CD19^cre* mice. **a** Flow cytometric detection of B220- and CD19-positive cells in spleens of indicated mice. Numbers represent the percent positive cells in the corresponding quadrant. **b** Presence of indicated immunoglobulin isotypes in the serum of indicated mice was quantified using enzyme-linked immunosorbent assay (ELISA) ($n = 4$ mice/genotype). **c** Six-week-old *Ufbp1^F/F* and *Ufbp1^F/F CD19^cre* mice were immunized with 4-hydroxy-3-nitrophenylacetyl conjugated to keyhole limpet hemocyanin (NP-KLH; 100 μg/mouse, intraperitoneally) in alum. NP-specific antibodies of IgM, IgG1, IgG2a, IgG2b, and IgG3 isotypes in the sera of the mice at indicated time points after immunization were quantified using ELISA ($n = 5$ mice/genotype). Samples were normalized against day 14 sera obtained from wild-type mice immunized with NP-KLH. **d** *Ufbp1^F/F* and *Ufbp1^F/F CD19^cre* mice were immunized with NP-Ficoll (25 μg/mouse) intraperitoneally in phosphate-buffered saline. NP-specific antibody of IgG3 isotype in the sera of mice at indicated time points after immunization was quantified as above ($n = 3$ mice/genotype). Error bars represent mean ± standard error. *$P < 0.05$, **$P < 0.01$, ***$P < 0.001$. Unpaired Student's two-tailed *t*-test was used. A representative of at least two experiments is shown

Ufbp1 and the genes involved in ufmylation pathway: Uba5, Ufc1, Ufl1, and Ufm1 by quantitative real-time PCR (qRT-PCR). These genes were expressed at a significantly higher level in plasma cells than naive B cells (Supplementary Fig. 2a). Bone marrow and spleens of *Ufbp1^F/F CD19^cre* mice contained significantly reduced frequency of plasma cells (BLIMP1+CD138^high and IRF4+Pax5−) than *Ufbp1^F/F* mice (Fig. 2a, b, Supplementary Fig. 2b, c). We also evaluated the frequency and number of plasma cells after immunization with NP-KLH. Following immunization, plasma cells were still present at a significantly lower frequency in bone marrow and spleens of *Ufbp1^F/F CD19^cre* mice than *Ufbp1^F/F* mice at all the time points evaluated (Fig. 2c, d, Supplementary Fig. 2d, e). ELISpot assay confirmed significantly lower numbers of ASCs in the bone marrow and spleens of *Ufbp1^F/F CD19^cre* mice than *Ufbp1^F/F* mice (Fig. 2e, f, Supplementary Fig. 2f, g). To analyze the development of antigen-specific plasma cells in vivo, mice were immunized with NP-KLP in the Sigma Adjuvant System (Monophosphoryl Lipid A-based adjuvant). One week later, 7-AAD−CD4−CD8−Gr-1−CD11b− cells were analyzed for binding to NP and plasma cells. Similar

fractions of IgD− cells showed binding to NP in the spleens of *Ufbp1^F/F CD19^cre* and *Ufbp1^F/F* mice, suggesting that clonal expansion of B cells following antigen encounter is unaffected by deficiency of Ufbp1 (Fig. 2g, h). However, significantly lower frequency of plasma cells (BLIMP1+CD138^high and BLIMP1+) among NP-specific cells was present in *Ufbp1^F/F CD19^cre* mice than *Ufbp1^F/F* mice (Fig. 2i, j). Collectively, these data demonstrate that Ufbp1 is dispensable for the expansion of B cells but plays an essential role in the differentiation of antigen-activated B cells into plasma cells.

**Differentiation of B cells into plasmablast depends on Ufbp1.** Naive B cells activated in vitro with LPS, interleukin (IL)-4, and IL-5 differentiate into plasmablasts. Like plasma cells, plasmablasts are ASCs, positive for CD138, BLIMP1, IRF4, negative for Pax5, and possess a highly developed ER network. In addition, the same set of transcription factors drive the development of plasmablasts and plasma cells[48]. Sorted naive B cells from the lymph nodes of *Ufbp1^F/F* and *Ufbp1^F/F CD19^cre*

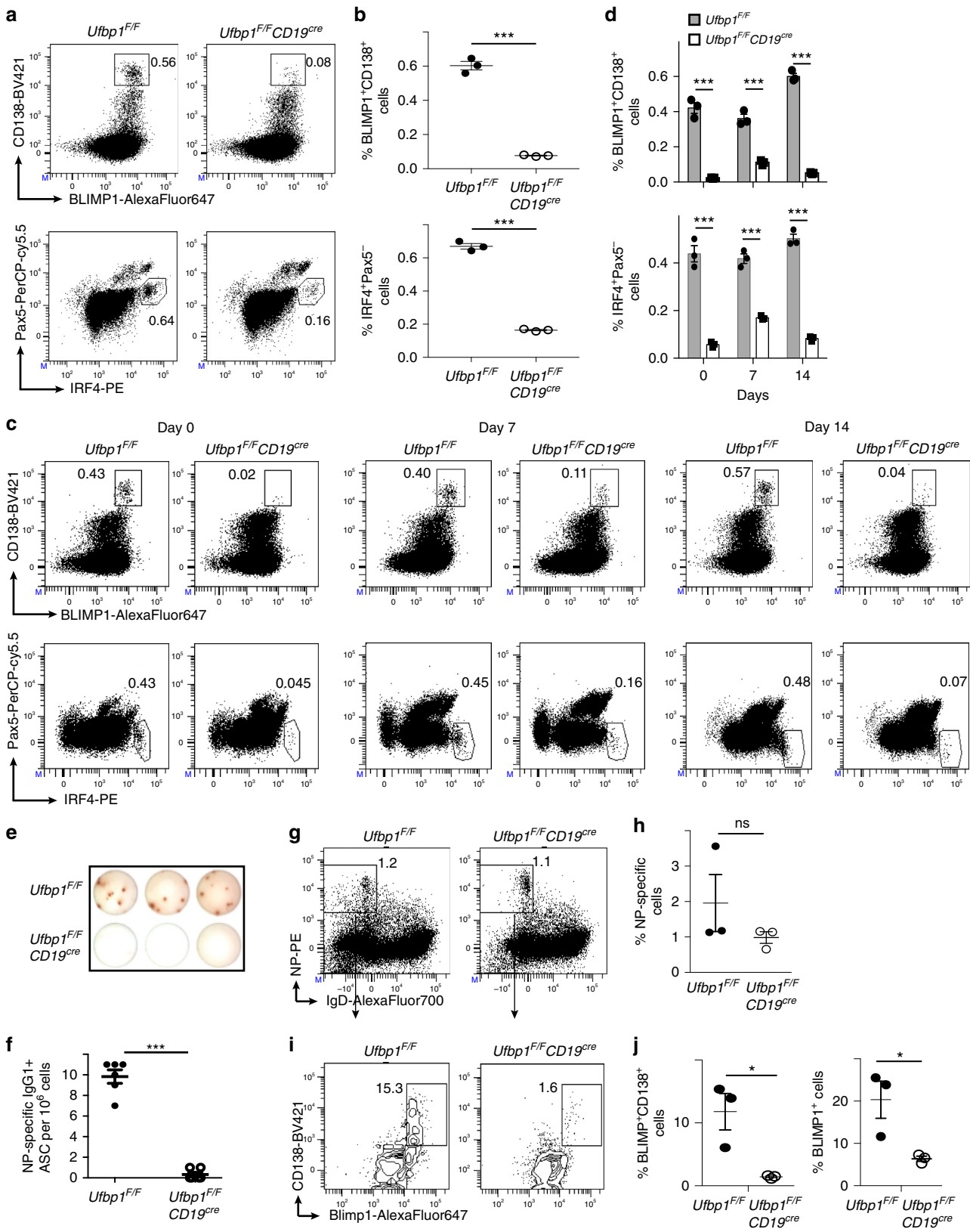

mice were cultured with LPS, IL-4, and IL-5. Four days later, the development of plasmablasts (BLIMP1+CD138high and IRF4+Pax5−) was analyzed in these cultures. Plasmablasts were present at significantly reduced frequency in the cultures started with B cells from *Ufbp1F/FCD19cre* mice than *Ufbp1F/F*

mice (Fig. 3a–d). Similarly, frequencies of single positive BLIMP1+ and IRF4+ cells were also significantly reduced in cultures started with B cells from *Ufbp1F/FCD19cre* than *Ufbp1F/F* mice (Supplementary Fig. 3a, b). Accordingly, the levels of secreted IgM were significantly decreased in the

**Fig. 2** Decreased number of plasma cells in *Ufbp1*[F/F]*CD19*[cre] mice. **a** Bone marrow cells from indicated mice were stained with BLIMP1, CD138, IRF4, and Pax5, and analyzed by flow cytometry. Shown is the BLIMP1 versus CD138 (upper panel), and IRF4 versus Pax5 staining (lower panel) **b** Enumeration of frequencies of BLIMP1+CD138+ and IRF4+Pax5− cells in **a** ($n = 3$ mice/genotype). **c** *Ufbp1*[F/F] and *Ufbp1*[F/F]*CD19*[cre] mice immunized with 4-hydroxy-3-nitrophenylacetyl conjugated to keyhole limpet hemocyanin (NP-KLH) as in Fig. 1c were analyzed for plasma cells (BLIMP1+CD138+ and IRF4+Pax5−) in bone marrow by flow cytometry at the indicated time points after immunization. **d** Enumeration of the frequencies of plasma cells (BLIMP1+CD138+ and IRF4+Pax5−) in **c** ($n = 3$ mice/genotype). **e** Indicated mice were immunized as in Fig. 1c. Twenty-one days later, bone marrow cells were analyzed for the presence of NP-specific IgG1+ASCs by ELISpot. Shown is a photograph of representative ELISpot wells. **f** Number (mean ± SEM) of NP-specific IgG1+ ELISpots in **e** ($n = 6$ mice/ genotype). **g** *Ufbp1*[F/F] and *Ufbp1*[F/F]*CD19*[cre] mice were immunized with NP-KLH in Sigma Adjuvant System and on day 10, 7-AAD−CD4−CD8−Gr-1−CD11b− splenic cells were analyzed for IgD and NP staining by flow cytometry. **h** Enumeration of the frequency of IgD−NP+ cells in **g** ($n = 3$ mice/genotype). **i** BLIMP1 and CD138 expression by IgD−NP+ cells in **g** was analyzed by flow cytometry. **j** Enumeration of frequency of BLIMP1+CD138[high] (left panel) and BLIMP1+ (right panel) cells among IgD−NP+ cells in **i** ($n = 3$ mice/genotype). Numbers represent the percent positive cells in the adjacent gated area. Error bars represent mean ± standard error. *$P < 0.05$, ***$P < 0.001$, ns nonsignificant. Unpaired Student's two-tailed *t*-test was used. A representative of at least two experiments is shown

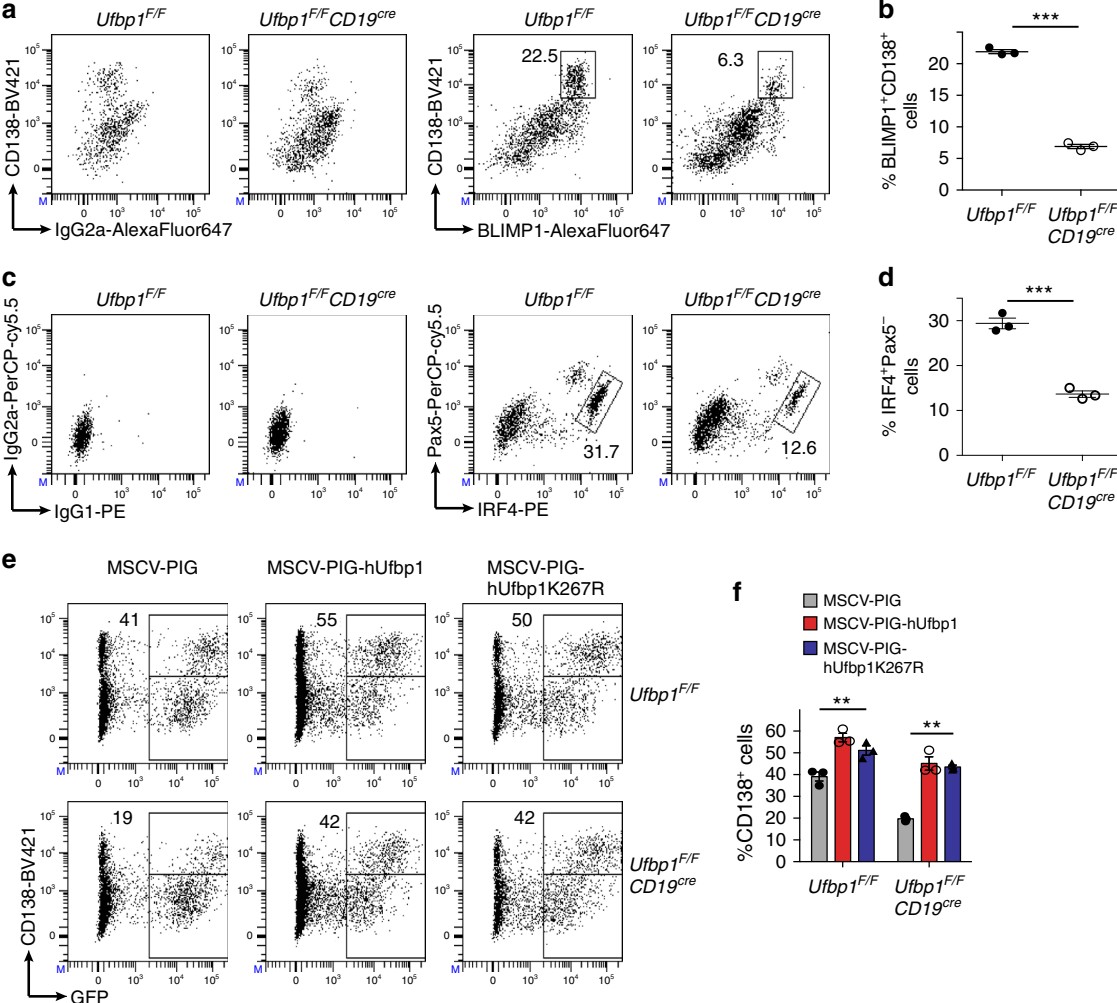

**Fig. 3** Defective differentiation of Ufbp1-deficient B cells into plasmablasts. **a**, **c** Sorted naive B cells from lymph nodes of indicated mice were cultured with lipopolysaccharide (LPS; 1 μg/ml) + interleukin (IL)-4 (10 ng/ml) + IL-5 (5 ng/ml). Four days later cells were analyzed for expression of BLIMP1, CD138, IRF4, and Pax5 by flow cytometry. Left panels in **a** and **c** show control IgG2a, IgG1, and IG2a isotype staining for BLIMP1, IRF4, and Pax5 respectively. Right panels in **a** and **c** show staining for BLIMP1 and CD138, and IRF4 and Pax5 respectively. Numbers represent the percent positive cells in the adjacent/ corresponding area. **b** Enumeration of the frequency of BLIMP1+CD138[high] cells in **a** ($n = 3$ mice/genotype). **d** Enumeration of the frequency of IRF4+Pax5− cells in **c** ($n = 3$ mice/genotype). **e** Naive B cells from *Ufbp1*[F/F] and *Ufbp1*[F/F]*CD19*[cre] mice were cultured with LPS, IL-4, and IL-5 as above, and transduced with retrovirus containing empty vector (MSCV-PIG), expressing human Ufbp1 (MSCV-PIG-hUfbp1) or Ufbp1K267R (MSCV-PIG-hUfbp1K267R). Transduction of cells with retroviruses was monitored by expression of green fluorescent protein (GFP). Two days after transduction, cells were analyzed for GFP and CD138. Numbers represent percent CD138+ cells within GFP+ cells. **f** Enumeration of the frequency of CD138+ cells among GFP+ cells in **e** ($n = 3$ mice/genotype). Error bars represent mean ± standard error. **$P < 0.01$, ***$P < 0.001$. Unpaired Student's two-tailed *t*-test was used. A representative of at least two experiments is shown

former cultures (Supplementary Fig. 3c). Most importantly, LPS-, IL-4-, and IL-5-induced defective development of Ufbp1-deficient B cells into plasmablasts was rescued by exogenous expression of Ufbp1, suggesting that Ufbp1 deficiency in B cells did not cause an irreversible defect in their ability to differentiate into plasmablasts (Fig. 3e, f, Supplementary Fig. 3d). Surprisingly, Ufbp1K267R, a mutant of Ufbp1 in which lysine at position 267 was changed to arginine and thus does not undergo ufmylation, promoted plasma cell development comparable to WT Ufbp1 (Fig. 3e, f, Supplementary Fig. 3d)[33]. Since lysine 267 (K267) is the main lysine in Ufbp1 that undergoes ufmylation, these data suggest that Ufbp1 promotes plasma cell development independent of its ufmylation at lysine 267. However, we cannot rule out the role of ufmylation of other conserved lysines at positions 116, 121, 124, 128, 193, 224, and 227 of Ufbp1 in development of plasma cells. Alternatively, it is possible that Ufbp1 promotes plasma cell development independent of its ufmylation. WT Ufbp1 significantly rescued immunoglobulin production by Ufbp1-deficient

plasmablasts, whereas Ufbp1K267R failed to do so (Supplementary Fig. 3e). Therefore, lysine 267 of Ufbp1 distinguishes between its role in plasma cell development versus immunoglobulin production. Collectively, these data demonstrate an essential role of Ufbp1 in the differentiation of B cells into plasmablasts.

**Ufbp1 expands ER in plasma cells.** To gain insight into the mechanism of Ufbp1-mediated regulation of plasma cell biology, we first analyzed the subcellular distribution of Ufbp1. Cell lysates from LPS-activated B cells were fractionated into cytosol and ER fractions and analyzed for Ufbp1. Figure 4a shows that Ufbp1 is exclusively present in the ER fraction. Immunocytochemistry staining confirmed co-localization of Ufbp1 with ER marker protein disulfide isomerase (PDI) (Fig. 4b). Due to Ufbp1 localization on the ER and a critical role for the ER in the functional biology of plasma cells, we analyzed the ER network in plasmablasts lacking Ufbp1. First, we determined the total ER mass in

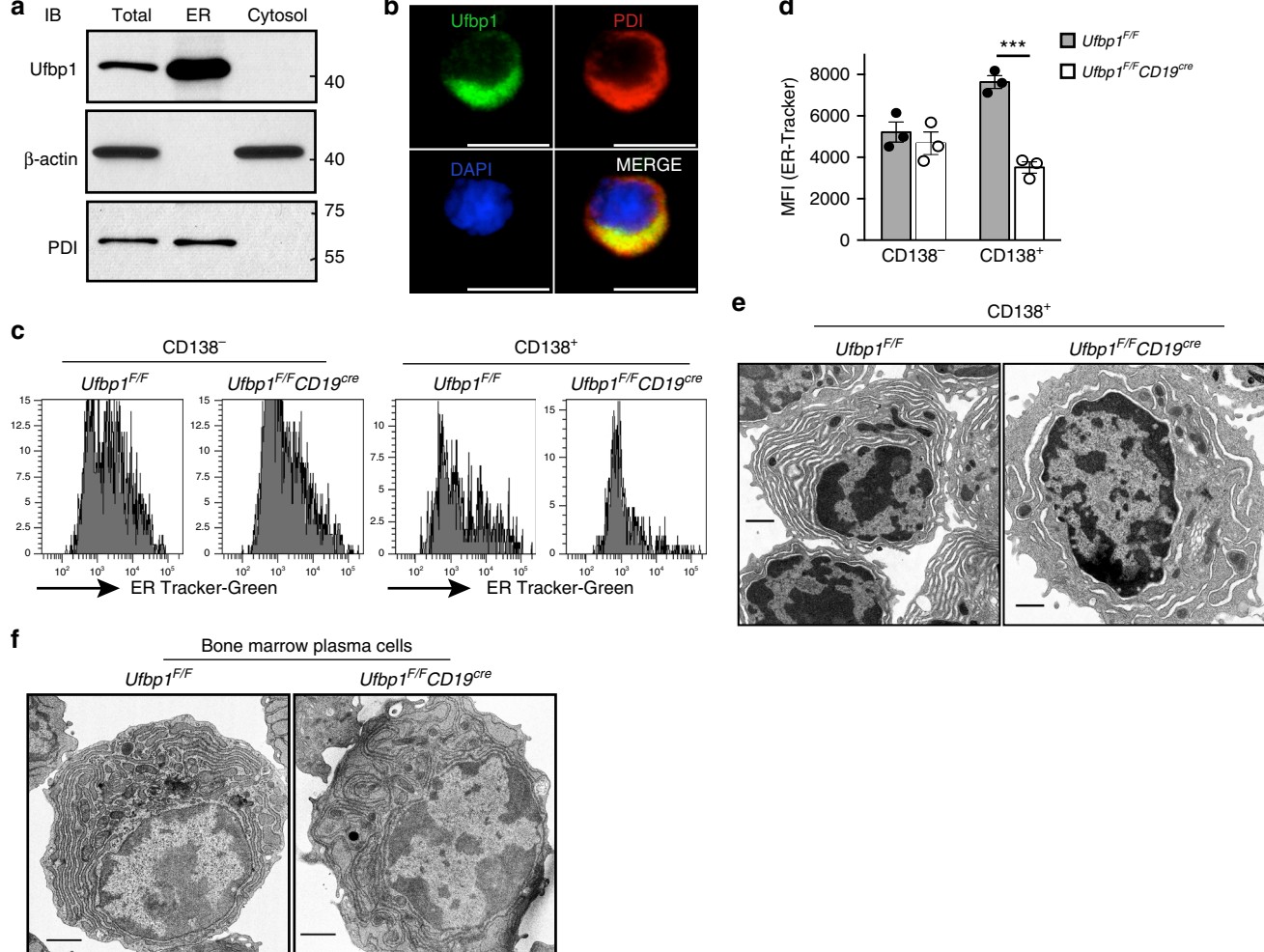

**Fig. 4** Ufbp1 is required for endoplasmic reticulum (ER) expansion in antibody-secreting cells. **a** Naive B cells from wild-type mice were activated with lipopolysaccharide (LPS) for 3 days. Total, ER, and cytoplasmic fractions were immunoblotted with antibodies against indicated molecules. PDI is used as ER marker, whereas β-actin is a marker for cytoplasm. **b** Visualization of Ufbp1 and PDI co-localization in LPS-activated B cells by fluorescent microcopy. Scale bars are 5 μm. **c** Naive B cells from Ufbp1[F/F] and Ufbp1[F/F]CD19[cre] mice were cultured with LPS for 3 days. Shown is the flow cytometric staining of ER-tracker in CD138⁻ and CD138⁺ cells. **d** Quantification of ER-tracker staining in CD138⁻ and CD138⁺ cells in **c** (n = 3 mice/genotype). MFI mean fluorescent intensity. **e** Transmission electron microscope images of CD138⁺ cells sorted from LPS-stimulated B cell cultures from Ufbp1[F/F] and Ufbp1[F/F]CD19[cre] mice. Scale bars are 1 μm. **f** Transmission electron microscope images of bone marrow plasma cells isolated from mice of indicated genotype. Scale bars are 1 μm. ***P < 0.001. Error bars represent mean ± standard error. Unpaired Student's two-tailed t-test was used. A representative of at least two experiments is shown

CD138$^-$ and CD138$^+$ cells from LPS-stimulated B cell cultures by an ER-tracker dye. On day 1 and 2 of LPS stimulation, Ufbp1-sufficient and -deficient B cells showed similar levels of ER staining (Supplementary Fig. 4a–d). On day 3 of LPS culture, both CD138$^-$ and CD138$^+$ cells are present in cultures. Ufbp1-sufficient and -deficient CD138$^-$ cells exhibited comparable staining with the ER-tracker (Fig. 4c, d). In contrast, Ufbp1-deficient CD138$^+$ cells showed significantly reduced staining for ER-tracker than their Ufbp1-sufficient counterparts, suggesting a lower ER mass in the former (Fig. 4c, d). Similarly, Ufbp1-deficient plasmablasts (CD138$^{high}$TACI$^+$) induced by LPS, IL-4, and IL-5 exhibited significantly decreased staining for ER-tracker dye than Ufbp1-sufficient counterparts (Supplementary Fig. 4e). We used transmission electron microscope (TEM) to analyze the ER network in Ufbp1-sufficient and -deficient CD138$^+$ cells. Ufbp1-sufficient CD138$^+$ cells displayed a well-developed ER network and most of the intracellular space was occupied by the ER, a characteristic of plasma cells (Fig. 4e). In sharp contrast, most of the intracellular space was devoid of the ER network in Ufbp1-deficient CD138$^+$ cells (Fig. 4e). Consistent with these findings and most importantly, TEM studies confirmed the absence of a well-developed ER network in bone marrow plasma cells from Ufbp1$^{F/F}$CD19$^{cre}$ mice, whereas the intracellular space of bone marrow plasma cells from Ufbp1$^{F/F}$ mice was filled with ER (Fig. 4f). We also evaluated ER-tracker staining on plasma cells from spleen and bone marrow. CD138 is not a reliable marker for identification for plasma cells in vivo. Fixation and permeabilization procedures used for staining of transcription factors associated with plasma cell identification may interfere with ER-tracker staining. Using BLIMP1-GFP mice, several groups have shown that CD138 in combination with TACI, Sca-1, or CD98 accurately identifies plasma cells in spleen and bone marrow[49–51]. Therefore, CD138$^{high}$TACI$^+$ cells (plasma cells) in bone marrow and spleens were analyzed for ER-tracker staining. Staining for the ER-tracker was significantly lower in plasma cells (CD138$^{high}$TACI$^+$) from the bone marrow and spleens of Ufbp1$^{F/F}$CD19$^{cre}$ mice than Ufbp1$^{F/F}$ mice (Supplementary Fig. 4f). Collectively, these data indicate that Ufbp1 plays an essential role in the expansion of the ER network in ASCs.

**Ufbp1 regulates IRE1α and PERK branches of UPR pathway.** XBP1 is a master transcription factor that plays an indispensable role in the ER expansion in plasma cells[16,25]. Due to the absence of a well-developed ER network in Ufbp1-deficient plasmablasts and plasma cells, the expression of XBP1 was analyzed. Surprisingly, XBP1 staining was significantly higher in bone marrow plasma cells from Ufbp1$^{F/F}$CD19$^{cre}$ mice than Ufbp1$^{F/F}$ mice (Fig. 5a, b). Similarly, CD138$^-$ and CD138$^+$ cells from LPS-stimulated B cell cultures from Ufbp1$^{F/F}$CD19$^{cre}$ mice showed significantly brighter staining for XBP1 than their counterparts from Ufbp1$^{F/F}$ mice (Supplementary Fig. 5a, b). It is unlikely that impaired transcriptional activity of XBP1 in Ufbp1-deficient plasma cells is the underlying mechanism for defective ER expansion, because the XBP1 target genes EDEM1, ERDJ4, PDI, and Sec61a were expressed at comparable levels in Ufbp1-sufficient and -deficient plasma cells (Supplementary Fig. 5c). Similarly, Ufbp1 deficiency did not affect the expression of Sec61a, PDI, and EDEM1 in CD138$^-$ and CD138$^+$ cells from LPS-stimulated cultures (Supplementary Fig. 5d). Induction of the UPR during development of plasma cells leads to activation of IRE1α. IRE1α-mediated splicing and subsequent activation of the transcription factor XBP1 plays an essential and indispensable role in the expansion of ER network[16,52]. Therefore, the role of Ufbp1 in regulation of UPR signaling was analyzed. Naive B cells from Ufbp1$^{F/F}$ and Ufbp1$^{F/F}$CD19$^{cre}$ mice were activated with LPS

and on day 0,1, 2, and 3 whole-cell lysates were immunoblotted with antibodies against molecules involved in UPR signaling. Expression of IRE1α protein increased in both Ufbp1-sufficient and -deficient LPS-activated B cells from day 0 to day 3 (Fig. 5c, d). Using short hairpin RNA-mediated knockdown of Ufbp1, a recent article shows that IRE1α is degraded in the absence of Ufbp1[53]. In contrast, we found that Ufbp1-deficient LPS-activated B cells expressed higher amount of total IRE1α protein than their Ufbp1-sufficient counterparts (Fig. 5c, d). Consistent with this, higher amount of spliced XBP1 mRNA and protein was present in Ufbp1-deficient than Ufbp1-sufficient LPS-activated B cells on day 1, 2, and 3 (Fig. 5c, d, Supplementary Fig. 5e, f). Similarly, Ufbp1 deficiency resulted in higher splicing of XBP1 in CD138$^-$ and CD138$^+$ cells from LPS-stimulated cultures and bone marrow plasma cells (Supplementary Fig. 5g, h). When expression of IRE1α and XBP1 was compared on day 1 and 2 between Ufbp1-sufficient and -deficient B cells, fold increase in expression of XBP1 in later correlated with higher expression of IRE1α in them (Fig. 5e), suggesting that at single-molecule level, Ufbp1 deficiency does not change the activity of IRE1α. Collectively, these data demonstrate that during B cell activation, absence of Ufbp1 leads to upregulation of IRE1α expression.

In addition to IRE1α activation, UPR also induces activation of PERK. The PERK branch of UPR is specifically suppressed in activated B cells[24,54]. On day 1 of LPS stimulation, phosphorylated form of PERK (p-PERK) was increased by ~>5-fold in Ufbp1-deficient B cells than Ufbp1-sufficient cells (Fig. 5f–h). A similar trend for phosphorylated form of eIF2α (p-eIF2α) was also observed (Fig. 5f–h). Consistent with this, protein level of ATF4 was increased by ~10-fold in Ufbp1-deficient than Ufbp1-sufficient B cells on day 1 (Fig. 5f–h). No noticeable difference in phosphorylation of PERK and eIF2α between Ufbp1-sufficient and -deficient B cells was observed on day 2 and 3. However, ATF4 was still expressed at higher level on day 2 and 3 in Ufbp1-deficient B cells, although the fold increase compared with Ufbp1-sufficient B cells was of lesser extent than on day 1. Activation of PERK during ER stress may lead to apoptosis[55]. Similar numbers of live cells were recovered at all the time points (day 1, 2, and 3) in cultures containing B cells from Ufbp1$^{F/F}$ and Ufbp1$^{F/F}$CD19$^{cre}$ mice (Supplementary Fig. 5i). Annexin-V and 7-AAD staining showed comparable survival of Ufbp1-sufficient and -deficient B cells in these cultures (Supplementary Fig. 5j). Taken together, these data suggest that Ufbp1 is a negative regulator of PERK activation.

**Ufbp1 promotes plasma cell development by suppressing PERK.** Next, we tested whether Ufbp1 deficiency-induced activation of PERK inhibits differentiation of Ufbp1-deficient B cells into plasma cells. To test this hypothesis, we generated mice lacking Ufbp1 and PERK in B cells, Perk$^{F/F}$Ufbp1$^{F/F}$CD19$^{cre}$. B cells from WT, Perk$^{F/F}$CD19$^{cre}$, Ufbp1$^{F/F}$CD19$^{cre}$, and Perk$^{F/F}$Ufbp1$^{F/F}$CD19$^{cre}$ mice were cultured with LPS and expression of ATF4 was analyzed as a marker of PERK activation. Deletion of PERK reduced the expression of ATF4 in the Ufbp1-deficient B cells to the levels observed in WT B cells (Supplementary Fig. 6a). Additionally, ATF4 level in WT B cells was comparable to that seen in PERK-deficient B cells. Next bone marrow and spleens of WT, Perk$^{F/F}$CD19$^{cre}$, Ufbp1$^{F/F}$CD19$^{cre}$, and Perk$^{F/F}$Ufbp1$^{F/F}$CD19$^{cre}$ mice were analyzed for plasma cells. Frequencies of plasma cells (BLIMP1$^+$CD138$^{high}$) in the spleens and bone marrow of WT, Perk$^{F/F}$CD19$^{cre}$, and Perk$^{F/F}$Ufbp1$^{F/F}$CD19$^{cre}$ mice were similar to each other and significantly higher than in Ufbp1$^{F/F}$CD19$^{cre}$ mice (Fig. 6a–d). These data suggested that deletion of PERK rescued development of Ufbp1-deficient B cells into plasma cells. To test this premise, B cells from WT, Perk$^{F/F}$CD19$^{cre}$, Ufbp1$^{F/F}$CD19$^{cre}$, and Perk$^{F/F}$Ufbp1$^{F/F}$CD19$^{cre}$ mice were cultured with LPS, IL-4, and IL-5, and their

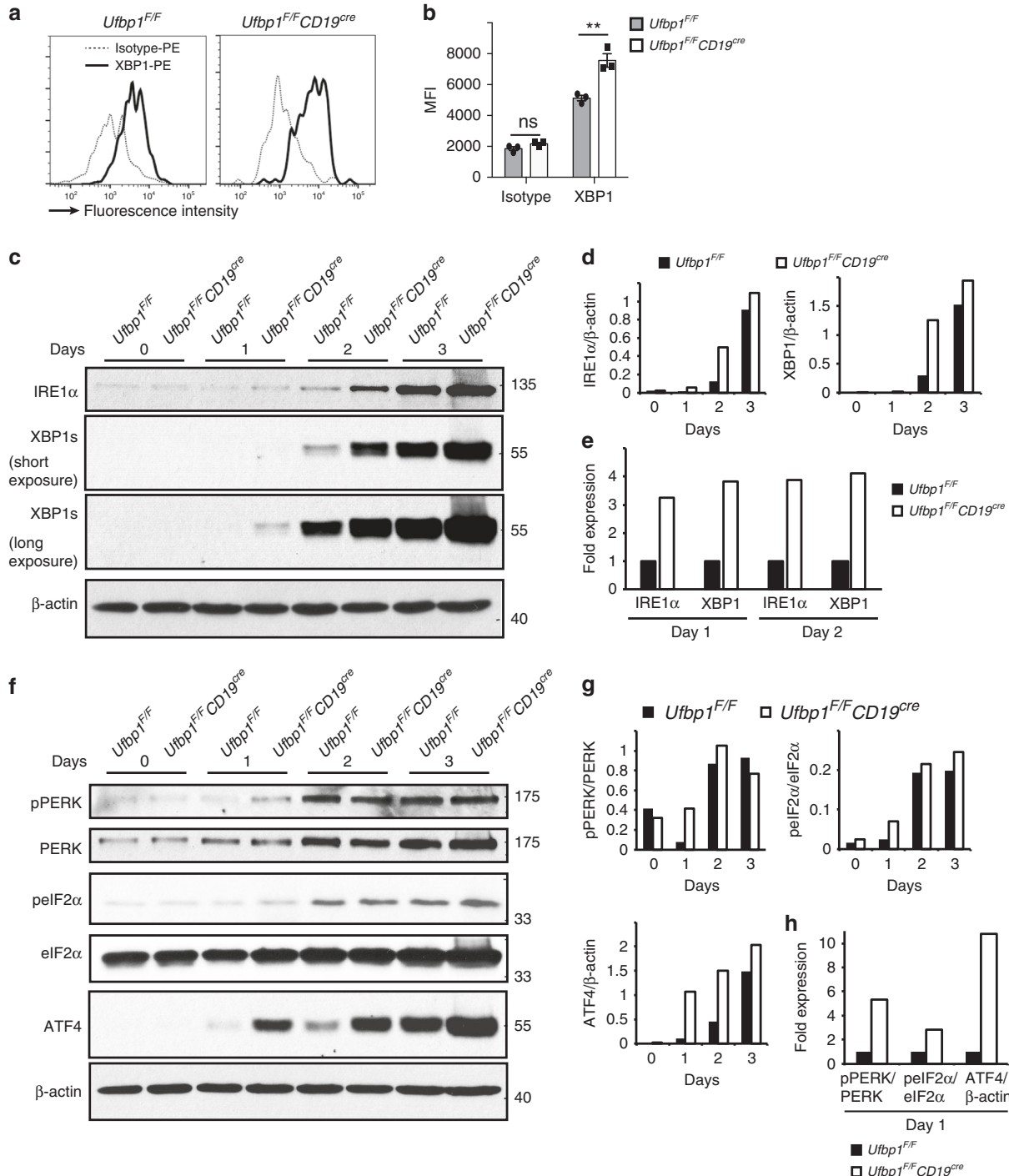

**Fig. 5** An essential role of Ufbp1 in regulation of unfolded protein response signaling. **a** Flow cytometric detection of spliced XBP1 (XBP1s) expression in bone marrow plasma cells of mice of indicted genotype. Dotted and solid lines represent the staining for control isotype and anti-XBP1s antibodies respectively. **b** Quantification of XBP1s expression in **a** ($n = 3$ mice/genotype). **c**, **f** Naive B cells from $Ufbp1^{F/F}$ and $Ufbp1^{F/F}CD19^{cre}$ mice were stimulated with lipopolysaccharide. At indicated time points, cells were harvested and total cell lysates were immunoblotted with antibodies against indicated molecules. **d**, **g** Quantification of expression of indicated molecules normalized against β-actin in **c** and **f** respectively. **e**, **h** Data on indicated days from **d** and **g** were replotted. Expression value of indicated molecules in cells from $Ufbp1^{F/F}$ mice on indicated time point was taken as 1. Error bars represent mean ± standard error. **$P < 0.01$. ns nonsignificant. Unpaired Student's two-tailed $t$-test was used. A representative of at least two experiments is shown

differentiation into plasmablasts was analyzed. Similar number of live cells were recovered in all the culture at all the time points, suggesting that deletion of PERK or Ufbp1 does not affect the survival or proliferation of B cells (Supplementary Fig. 6c). On day 4, comparable frequency of plasmablasts (BLIMP1⁺CD138^high and IRF4⁺Pax5⁻ cells) developed in cultures of WT and PERK-

deficient B cells, which was significantly higher than in cultures of Ufbp1-deficient B cells (Fig. 6e, f, Supplementary Fig. 6d). Deletion of PERK rescued the defective development of Ufbp1-deficient B cells into plasmablasts, because frequencies of BLIMP1⁺CD138^high and IRF4⁺Pax5⁻ cells in cultures of B cells lacking both Ufbp1 and PERK were significantly higher than in cultures of B cells lacking

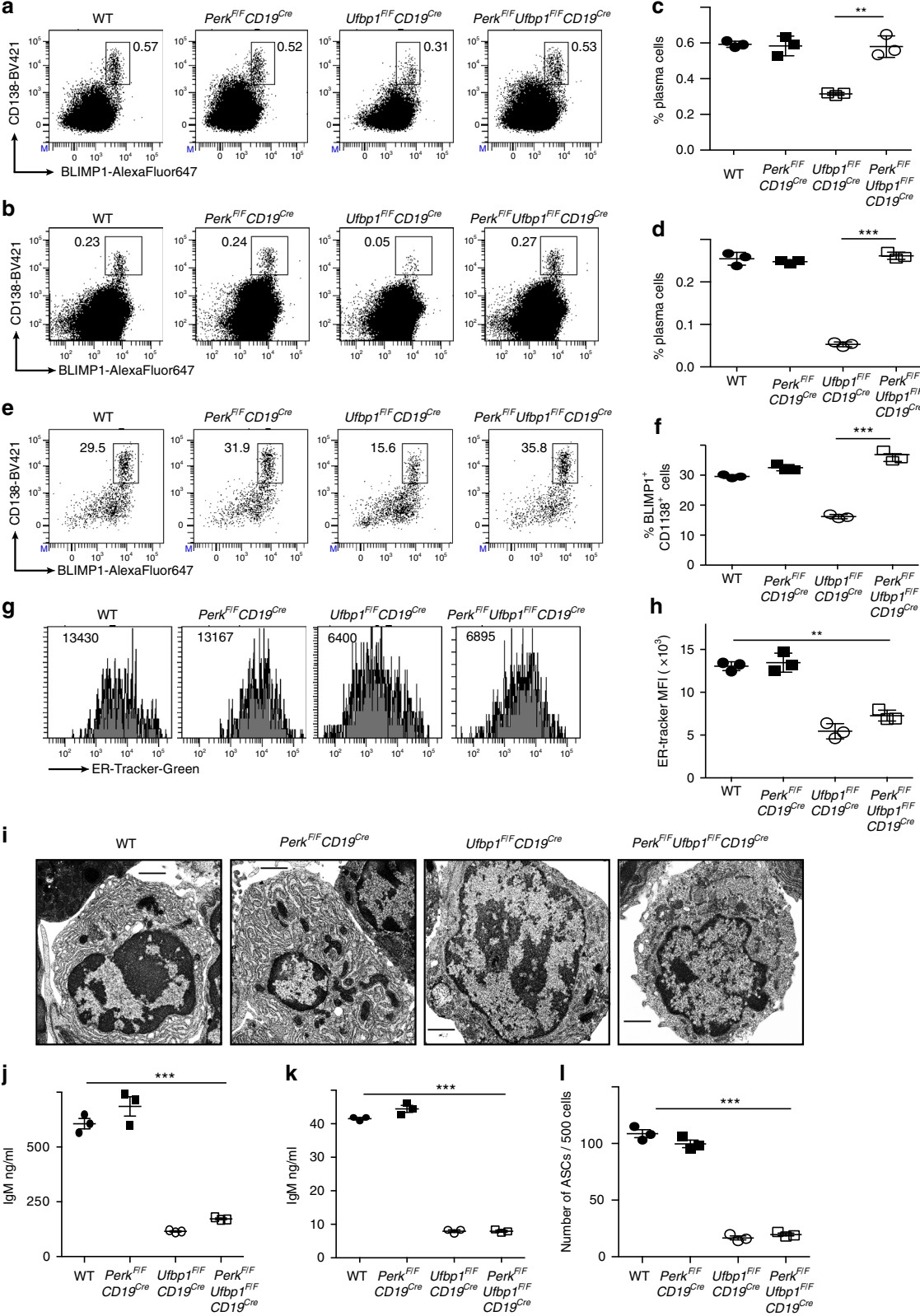

Ufbp1 (Fig. 6e, f, Supplementary Fig. 6d). Taken together, these data demonstrate that Ufbp1 promotes plasma cell development by suppressing PERK activation.

We also tested whether deletion of PERK restores the defective ER expansion in Ufbp1-deficient plasma cells and plasmablasts. Spleen and bone marrow plasma cells lacking either Ufbp1 or both

Ufbp1 and PERK showed significantly decreased staining for ER-tracker than their WT or PERK-deficient counterparts (Fig. 6g, h, Supplementary Fig. 6b). Similarly, WT and PERK-deficient plasmablasts (CD138highTACI+ cells) induced by LPS, IL-4, and IL-5 showed significantly higher staining for ER-tracker than those lacking either Ufbp1 or both Ufbp1 and PERK (Supplementary

**Fig. 6** Ufbp1 promotes development of plasma cells by suppressing PKR-like ER protein kinase (PERK). **a**, **b** BLIMP1 and CD138 expression by splenocytes (**a**) and bone marrow cells (**b**) from indicated mice was analyzed by flow cytometry ($n = 3$ mice/genotype). **c**, **d** Enumeration of frequency of plasma cells (BLIMP1$^+$CD138$^{high}$) in **a** and **b** respectively ($n = 3$ mice/genotype). **e** Naive B cells from indicated mice were stimulated with lipopolysaccharide (LPS), interleukin (IL)-4, and IL-5 as in Fig. 3a. Four days later expression of BLIMP1 and CD138 was analyzed by flow cytometry. **f** Enumeration of frequency of BLIMP1$^+$CD138$^{high}$ cells in **e** ($n = 3$ mice/genotype). **g** ER-tracker staining of splenic plasma cells (CD138$^+$TACI$^+$ cells) from indicated mice. **h** Quantification of ER-tracker staining of plasma cells in **g** ($n = 3$ mice/genotype). **i** Transmission electron microscope images of CD138$^+$ cells sorted from LPS-stimulated B cell cultures from indicated mice. Scale bars are 1 μm. **j** CD138$^+$ cells ($10^4$ cells/well) sorted from LPS-stimulated B cell cultures from indicated mice were cultured for 4 h and IgM production in culture supernatants was analyzed by enzyme-linked immunosorbent assay (ELISA). **k** Bone marrow plasma cells ($4 \times 10^3$ cells/well) from indicated mice were cultured overnight and IgM production in culture supernatants was analyzed by ELISA. **l** Enumeration of antibody-secreting cells by ELISpot in CD138$^+$ population sorted from LPS-stimulated B cell cultures from indicated mice. Error bars represent mean ± standard error. **$P < 0.01$, ***$P < 0.001$. Unpaired Student's two-tailed $t$-test was used. A representative of two experiments is shown

Fig. 6e). Consistent with this, TEM images confirmed absence of a well-developed ER network in plasmablasts from LPS-stimulated B cell cultures from $Ufbp1^{F/F}CD19^{cre}$ and $Perk^{F/F}Ufbp1^{F/F}CD19^{cre}$ mice (Fig. 6i). On the other hand, plasmablasts from LPS-stimulated B cell cultures from both WT and $Perk^{F/F}CD19^{cre}$ mice possessed a highly developed ER network (Fig. 6i). A well-developed ER network plays an important role in immunoglobulin secretion by plasma cells. Therefore, immunoglobulin secretion by plasma cells and plasmablasts from WT, $Perk^{F/F}CD19^{cre}$, $Ufbp1^{F/F}CD19^{cre}$, and $Perk^{F/F}Ufbp1^{F/F}CD19^{cre}$ mice were analyzed. Bone marrow plasma cells and plasmablasts from $Ufbp1^{F/F}CD19^{cre}$ and $Perk^{F/F}Ufbp1^{F/F}CD19^{cre}$ mice produced significantly less IgM antibody than their counterparts from WT and $Perk^{F/F}CD19^{cre}$ mice (Fig. 6j, k). ELISpot assay also showed significantly decreased number of ASCs in plasmablasts from $Ufbp1^{F/F}CD19^{cre}$ and $Perk^{F/F}Ufbp1^{F/F}CD19^{cre}$ mice than WT and $Perk^{F/F}CD19^{cre}$ mice (Fig. 6l, Supplementary Fig. 6f). Accordingly, and notwithstanding with normal numbers of plasma cells in bone marrow and spleen, $Perk^{F/F}Ufbp1^{F/F}CD19^{cre}$ mice exhibited significantly reduced levels of all the immunoglobulin isotypes in their sera than WT and $Perk^{F/F}CD19^{cre}$ (Supplementary Fig. 6g). Collectively, these data demonstrate that Ufbp1 promotes plasma cell development via suppressing PERK branch of UPR but stimulates ER homeostasis and plasma cell function in a PERK-independent manner.

**Ufbp1 downstream to XBP1 expands the ER network**. Notwithstanding with the higher expression of XBP1, Ufbp1-deficient plasma cells and plasmablasts show a defective ER expansion. Therefore, we hypothesized that Ufbp1 and the ufmylation system are induced by the IRE1α/XBP1 axis in plasma cells and plasmablasts. $Ire1\alpha^{F/F}$ and $Ire1\alpha^{F/F}CD19^{cre}$ mice were utilized to address this hypothesis. Expression of $Ire1\alpha$ mRNA was drastically reduced in naive B cells and bone marrow plasma cells of $Ire1\alpha^{F/F}CD19^{cre}$ mice than $Ire1\alpha^{F/F}$ mice suggesting efficient deletion of $Ire1\alpha$ gene in the former mice (Fig. 7a). We found that expression of Ufbp1 as well all the molecules involved in ufmylation pathway Uba5, Ufc1, Ufl1, and Ufm1 was significantly lower in bone marrow plasma cells from $Ire1\alpha^{F/F}CD19^{cre}$ mice than $Ire1\alpha^{F/F}$ mice (Fig. 7a). The deficiency was specific to the plasma cells, because IRE1α-sufficient and -deficient naive B cells express comparable levels of Ufbp1, Uba5, Ufc1, Ufl1, and Ufm1 (Fig. 7a, b, Supplementary Fig. 7a). LPS, a known inducer of differentiation of naive B cells into plasmablasts, significantly upregulated the expression of Ufbp1, Uba5, Ufc1, Ufl1, and Ufm1 in WT B cells (Fig. 7b, Supplementary Fig. 7a). In contrast, under similar conditions, LPS failed to upregulate Ufbp1, Uba5, Ufc1, Ufl1, and Ufm1 in IRE1α-deficient B cells (Fig. 7b, Supplementary Fig. 7a). At protein level, expression of ~48%, 34%, 52%, 43%, and 60% Ufbp1, Uba5, Ufc1, Ufl1, and Ufm1 respectively is IRE1α/XBP1-independent in LPS-activated B cells (Supplementary Fig. 7a). A similar data were observed when the expression of

Ufbp1, Uba5, Ufc1, Ufl1, and Ufm1 was compared between IRE1α-sufficient and -deficient CD138$^-$ and CD138$^+$ cells sorted from LPS-stimulated B-cell cultures (Fig. 7c). Whole-cell lysates from LPS-activated B cells from WT, $Ire1\alpha^{F/F}CD19^{cre}$, and $Ufbp1^{F/F}CD19^{cre}$ mice were immunoblotted with anti-Ufm1 antibody to gain insight into ufmylation activity of these cells. Several ufmylated bands were present at lower levels in LPS-activated B cells from $Ire1\alpha^{F/F}CD19^{cre}$ mice than WT B cells (Supplementary Fig. 7b). Interestingly, intensity of ufmylated bands of approximate molecular weight of 30 and 35 kDa was reduced in IRE1α-deficient B cells, whereas intensity of 30 kDa band was unchanged and of 35kDa was increased in Ufbp1-deficient B cells than WT counterparts (Supplementary Fig. 7b). This is not surprising because in Ufbp1-deficient B cells either ufmylation events downstream to Ufbp1 are expected to be reduced and/or in the absence of Ufbp1, ufmylation of Ufbp1-independent targets may go up. In contrast, due to decreased expression of all the molecules involved in ufmylation pathway, ufmylation of all the targets are estimated to go down in IRE1α-deficient B cells. Consistent with the essential role of IRE1 in the activation of transcription factor XBP1, LPS-activated B cells from $Ire1\alpha^{F/F}$ mice expressed transcription factor XBP1, whereas XBP1 expression was undetectable in LPS-activated B cells from $Ire1\alpha^{F/F}CD19^{cre}$ mice (Fig. 7b). Next, LPS-stimulated WT and IRE1α-deficient B cell cultures were transduced with retroviruses expressing empty vector or unspliced or spliced XBP1. Figure 7d and Supplementary Fig. 7c show that ectopic expression of spliced XBP1 (XBP1s) but not unspliced XBP1 (XBP1u) in IRE1α-deficient CD138$^+$ cells from LPS-stimulated cultures significantly upregulated the expression of Ufbp1, Uba5, Ufc1, Ufl1, and Ufm1. These findings suggest that Uba5, Ufc1, Ufl1, Ufbp1, and Ufm1 are expressed in plasma cells and LPS-activated B cells in both IRE1α/XBP1-dependent and -independent manner. Upregulation of all the components of Ufm1 system: Uba5, Ufc1, Ufl1, and Ufm1 along with Ufbp1 by IRE1α/XBP1 and defective ER expansion in CD138$^+$ cells lacking IRE1α, XBP1, or Ufbp1 suggest that ufmylation of Ufbp1 may play an important role in ER expansion. To test this hypothesis, IRE1α-deficient B cells were activated with LPS, IL-4, and IL-5 and transduced with retroviruses expressing empty vector and Ufbp1. As expected from previous studies, IRE1α-deficient plasmablasts from empty retrovirus-infected cells showed significantly reduced staining for ER-tracker than WT counterpart, suggesting decreased ER mass in former (Fig. 7e). Overexpression of Ufbp1 partially but significantly rescued the ER mass in IRE1α-deficient plasmablasts (Fig. 7e). The partial expansion of ER in Ufbp1-expressing IRE1α-deficient plasmablasts may undermine the role of Ufbp1 in ER expansion because all the enzymes required for ufmylation of Ufbp1 are expressed at much lower level in these cells (Fig. 7b, c), which may result in under-ufmylation of Ufbp1. Consistent with this, Ufbp1K267R expression did not significantly change the ER staining in IRE1α-deficient plasmablasts. Since ER

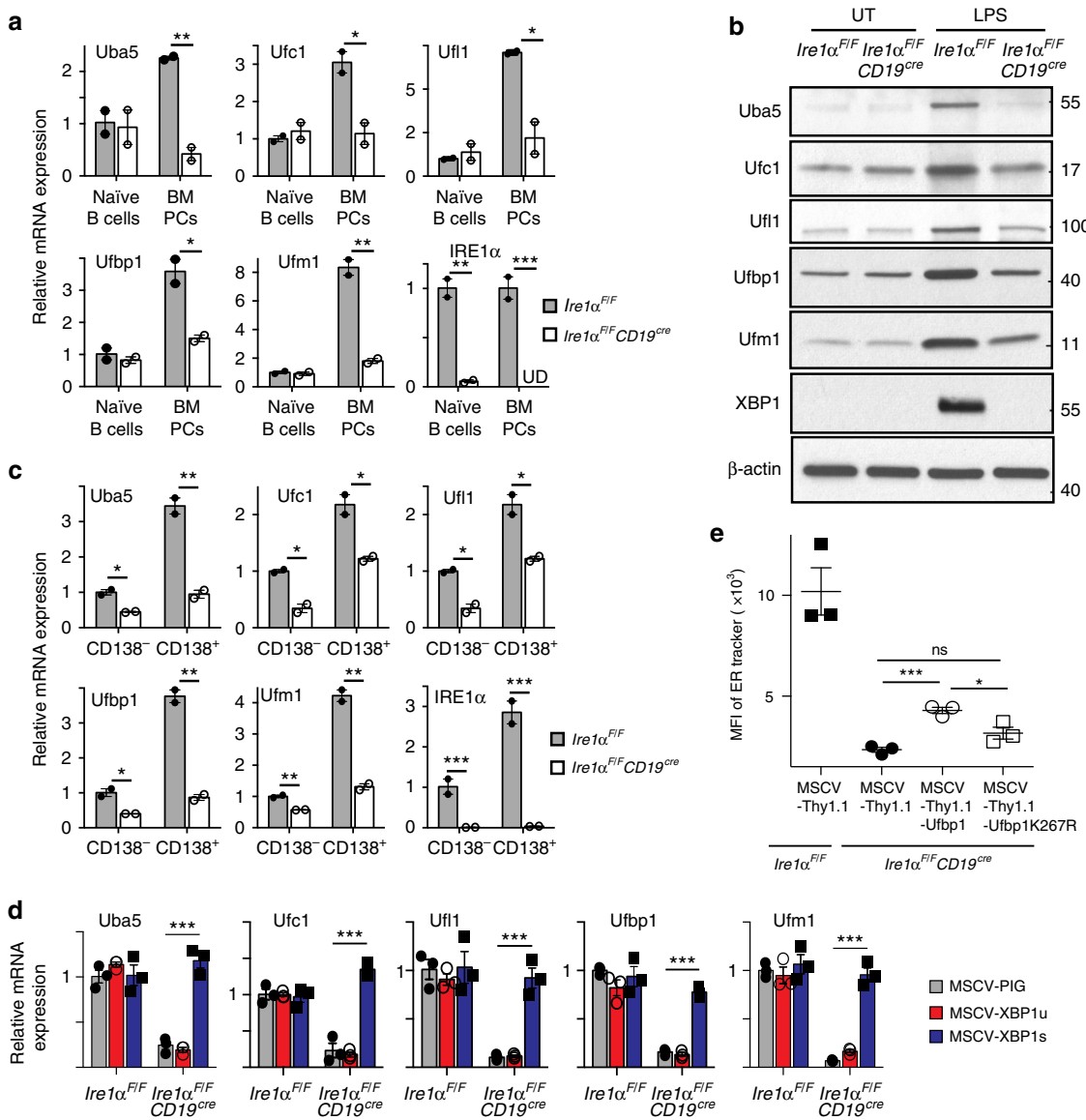

**Fig. 7** Ufbp1 downstream to IRE1/XBP1 induces endoplasmice reticulum (ER) expansion. **a** Naïve B cells and plasma cells were sorted from the spleens and bone marrows respectively of *Ire1α^F/F* and *Ire1α^F/F CD19^cre* mice. Expression of the indicated molecules normalized against β-actin was quantified by quantitative real-time PCR (qRT-PCR). **b** Total cell lysates from naïve B cells (untreated) and treated with lipopolysaccharide (LPS) for 3 days from indicated mice were immunoblotted with antibodies against indicated molecules. **c** Naïve B cells from *Ire1α^F/F* and *Ire1α^F/F CD19^cre* mice were treated with LPS. Three days later cells were sorted into CD138− and CD138+ populations. Expression of Uba5, Ufc1, Ufl1, Ufbp1, Ufm1, and IRE1α mRNA normalized against β-actin was quantified by qRT-PCR. **d** Naïve B cells from *Ire1α^F/F* and *Ire1α^F/F CD19^cre* mice were stimulated with LPS and transduced with retroviruses expressing empty vector (MSCV-PIG), unspliced XBP1 (MSCV-XBP1u), or spliced XBP1 (MSCV-XBP1s) on day 2. On day 4, CD138+ cells transduced with viruses were sorted and expression of Uba5, Ufc1, Ufl1, Ufbp1, and Ufm1 mRNA normalized against β-actin was quantified by qRT-PCR. Error bars represent standard error of mean. **e** LPS-stimulated B cells from wild-type and *Ire1α^F/F CD19^cre* mice were transduced with empty retrovirus (MSCV-Thy1.1), retrovirus expressing Ufbp1 or Ufbp1K267R. Cells were stained with anti-Thy1.1, CD138, and ER-tracker dye, and analyzed by flow cytometry. Shown is the mean fluorescent intensity (MFI) of the ER-tracker staining on the CD138+ cells among cells transduced with retrovirus (Thy1.1+ cells). Error bars represent mean ± standard error. *P < 0.05, **P < 0.01, ***P < 0.001, ns nonsignificant. Unpaired Student's two-tailed *t*-test was used. A representative of at least two experiments is shown

expansion plays a critical role in immunoglobulin production[16,25,56], inability of Ufbp1K267 to expand ER is consistent with its failure to enhance production of IgM by Ufbp1-deficient plasmablasts in Supplementary Fig. 3e. The partial rescue of ER mass in Ufbp1-expressing IRE1α-deficient plasmablasts may also be due to lower expression of other XBP1 targets (such as PDI, EDEM1, ERDj4, Sec61a, etc) involved in ER expansion. Taken together, these data suggest that although expression of Ufbp1 and components of ufmylation system in naïve B cells are

IRE1α/XBP1-independent, their upregulation downstream of IRE1α/XBP1 plays a critical role in ER expansion in plasma cells and plasmablasts.

## Discussion

In this report, we show that Ufbp1, a target and co-factor of the Ufm1 system plays an essential role in the development of plasma cells and expansion of the ER network. Previous studies have

reported specific suppression of PERK pathway in developing plasma cells[23,24]. Our study shows that Ufbp1 plays an essential role in suppression of PERK pathway. The relevance of this suppression is evident from the finding that deletion of PERK restored the defective development of Ufbp1-deficient B cells into plasma cells. Expression of Ufbp1 and other molecules involved in ufmylation pathway in naive B cells is independent of IRE1α/XBP1. In contrast, in plasma cells, expression of a significant fraction of Ufbp1 downstream to IRE1α/XBP1 performs a critical function in expansion of ER network. Thus, differential involvement of Ufbp1 in regulation of different steps of UPR pathway has distinct outcomes on the development and function of plasma cells.

Skin fibroblasts from individuals with biallelic mutations in Uba5, one coding for a truncated protein and the other coding for a loss of function resulting in a defective ufmylation pathway, are resistant to tunicamycin-induced toxicity and show an expanded ER network[35]. In contrast, our findings show that Ufbp1 deficiency leads to impaired expansion of the ER network. Although it is possible that the ufmylation pathway may differentially affect the ER network in different cell types, the opposing effects may also be explained by the fact that the loss of Uba5 function is expected to result in no-ufmylation of all the targets, whereas Ufbp1 deficiency will only affect the ufmylation of molecule(s) downstream to it. Therefore, a better understanding of the ufmylation pathway and its targets are needed to understand the importance of this pathway in the regulation of ER homeostasis in plasma and other secretory cells.

In contrast to numerous studies focusing on transcriptional regulation of gene expression, only a handful of studies have aimed to address the control of mRNA translation in regulating cellular differentiation and function. mTOR signaling regulates translation of mRNA involved in the biology of prostate cancer proliferation, metabolism, and invasion[57]. Ribosome profiling has demonstrated how translational control of mRNA regulates cell cycle[58,59]. Ribosome-associated protein RPL38 plays a key role in the translation of a selected set of homeobox mRNAs and thereby regulates tissue patterning[60]. A recent article showed that Ufl1 is part of the ribosome and ufmylates ribosome-associated proteins uS3, uS10, uL16, and eIF6[61]. Given the strong association between Ufl1 and Ufbp1[33,44], it is possible that the Ufl1/Ufbp1 complex is part of the ribosomes and regulate their composition and/or function on the ER. Future studies are necessary to test (1) whether Ufl1/Ufbp1 complex regulates translational control during plasma cell development, and (2) if so, its relevance to the biology of plasma cells.

The current study also shows that Ufbp1 regulates development of plasma cells by suppression of PERK pathway. Accumulation of ATF4 is the first stable product of the activation of PERK pathway. Higher level of ATF4 was present in LPS-activated Ufbp1-deficient B cells than WT B cells. In contrast, ATF4 level between WT, PERK-deficient, and PERK/Ufbp1 double-deficient B cells was comparable. These findings suggest that Ufbp1 completely suppresses PERK activation during development of plasma cells and may explain the previous findings showing no abnormality in development of plasma cells in mice lacking PERK. How does Ufbp1 suppress activation of PERK? A recent article shows that Ufbp1 is required for stability of IRE1α and thus depletion of Ufbp1 results in diminished IRE1α/XBP1 signaling leading to disturbed ER homeostasis and activation of PERK[53]. However, higher expression and activation of IRE1α/XBP1 in Ufbp1-deficient B cells argues against this possibility. Alternatively, it is possible that Ufbp1 represses PERK activity by directly interacting with it or indirectly by influencing some critical step in ER homeostasis. PERK is known to regulate survival, self-renewal, proliferation, and differentiation of

different cell types such as osteoblasts, skeletal muscle, adipocytes, neonatal pancreatic β-cells, and hematopoietic stem cells[62–65]. Its effect on proliferation and differentiation of insulin-secreting β-cells of endocrine pancreas during fetal and neonatal development is unrelated to ER stress and UPR pathway[63]. Additionally, PERK promotes adipocyte differentiation via intrinsic lipid kinase activity[66]. Therefore, the possibility arises that PERK may suppress the differentiation of naive B cells into plasma cells via pathway not related to ER stress or UPR pathway. In summary, the current study shows an unprecedented role for Ufbp1 and the ufmylation pathway in the development and function of plasma cells by differentially affecting the different steps of UPR pathway and thus suggests that manipulation of ufmylation pathway may have therapeutic potential for the treatment of plasma cell-related pathologies.

## Methods

**Mice**. $Ufbp1^{F/F}$ mice and $Ire1\alpha^{F/F}$ mice were previously described[44,67]. C57BL/6J, $Perk^{F/F}$, and $CD19^{Cre}$ mice were purchased from The Jackson Laboratory. $Ufbp1^{F/F}$, $Perk^{F/F}$, and $Ire1\alpha^{F/F}$ mice were crossed with $CD19^{Cre}$ mice to obtain $Ufbp1^{F/F}CD19^{cre}$, $Perk^{F/F}CD19^{cre}$ and $Ire1\alpha^{F/F}CD19^{cre}$ mice respectively. $Ufbp1^{F/F}CD19^{cre}$ and $Perk^{F/F}$ mice were used to generate $Perk^{F/F}Ufbp1^{F/F}CD19^{cre}$ mice. The following PCR primers were used for genotyping of the indicated mice: $Ufbp1^{F/F}$ (5′-TAGTACTTGAAGTCTGGCTTGGTA-3′ and 5′-TAGTCAGGAACTGATG AGTGTCTC-3′), $Ire1\alpha^{F/F}$ (5′-CAGAGATGCTGAGTGAAGAC-3′ and 5′-ACA GTGGTTCCTGTGAAGGT-3′), $CD19^{Cre}$ (5′-GTGAAACAGCATTGCTGTCACT T-3′, 5′-TGGTCTGAGACATTGACAATCA-3′, and 5′-CTGGCTACCATGCCAT CTCC-3′), and $Perk^{F/F}$ (5′-TTGCACTCTGGCTTTCACTC-3′ and 5′-AGGAGGA AGGTGGAATTTGG-3′). The Institutional Animal Care and Use Committee, Augusta University approved all the animal procedures. Mice of both sexes between 6 and 24 weeks of age were randomly used.

**Immunizations**. To study the TD antibody responses, 6-week-old $Ufbp1^{F/F}$ CD19Cre and $Ufbp1^{F/F}$ mice were immunized i.p. with NP-KLH (100 μg/mouse, Bioresearch Technologies, cat # N5060) mixed with alum adjuvant (ThermoFisher, cat # 77161). Mice were bled at time of immunization and day 7, 21, and 90. For TI antibody responses, NP-Ficoll (25 μg/mouse, Bioresearch Technologies, cat # F1420) dissolved in phosphate-buffered saline (PBS) was injected i.p and presence of NP-specific antibody in the serum was analyzed on day 21. For NP-specific plasma cell responses, mice were immunized i.p. with NP-KLH (400 μg/mouse) emulsified in Sigma Adjuvant System (Sigma-Aldrich, cat # S6322).

**B cell isolation and culture in vitro**. B cells were isolated from lymph nodes as B220-positive cells by using B220 microbeads (Miltenyi Biotec, cat # 130-049-501). Purified B cells were cultured at a density of $0.5 \times 10^6$ cells per 2 ml per well in 24-well plate or $0.1 \times 10^6$ cells per 0.2 ml well in 96-well plate in RPMI1640 medium supplemented with penicillin/streptomycin (Corning, cat # 30-002-CI), glutamine (Corning, cat # 25-005-CI), 10 mM Hepes (Gibco, cat # 15630-080), 50 μM β-mercaptoethanol (Fisher Scientific, cat # CAS 60-24-2), and 10% fetal bovine serum (FBS; Corning, cat # 35-010-CV) containing 25 μg/ml LPS (Sigma-Aldrich, cat # L4130) or LPS (1 μg/ml) + rmIL-4 (Peprotech, cat # 214-14, 10 ng/ml) + rmIL-5 (Biolegend, cat # 581502, 5 ng/ml). At indicated time points, cultures were harvested and used in experiments as mentioned in figure legends.

**Flow cytometry and cell sorting**. Single-cell suspensions from lymph nodes, spleen, or bone marrow were suspended in PBS containing 1% FBS and 10 mM Hepes, and stained with antibodies against following surface molecules: CD19-PerCP-CY5.5 (clone: 6D5, Biolegend, cat # 115534, 1:400), B220-PE-Cy7 (clone: RA3-6B2, Biolegend, cat # 103222, 1:400), B220-APC (clone: RA3-6B2, Biolegend, cat # 553092, 1:400), CD138-BV421 (clone: 281-2, BD Biosciences, cat # 562610, 1:500), CD138-PE (clone: 281-2, BD Biosciences, cat # 561070, 1:500), IgD-AlexaFluor700 (clone: 11-26c.2a, Biolegend, cat # 405729, 1:400), IgD-FITC (clone: 11-26c.2a, BD Biosciences, cat # 553439, 1:400), IgM-BV786 (clone: R6-60.2, BD Biosciences, cat # 564028, 1:400), Gr1-PerCP-CY5.5 (clone: RB6-8C5, BD Biosciences, cat # 561103, 1:400), CD4-PerCP-CY5.5 (clone: RM4-5, ThermoFisher, cat # 45-0042-80, 1:400), CD8-PerCP-CY5.5 (clone: 53-6.7, ThermoFisher, cat # 45-0081-80, 1:400), CD11b-PerCP-CY5.5 (clone: M1/70, ThermoFisher, cat # 45-0112-80, 1:400), and TACI-PE (Clone:8F10, Biolegend, cat # 133403, 1:400). In some experiments, cells were also stained with NP-PE (Bioresearch Technologies, cat # N5070-1, 1:10000) and 7-AAD (ThermoFisher, cat # 006993-50, 1:75). ER-Tracker green (ThermoFisher, cat # E34251, 1:1000), ER-Tracker red (ThermoFisher, cat # E34250, 1:1000), and annexin-V-PE (ThermoFisher, cat # 12-8102-69, 1:75) staining was performed according to the manufacturer's instructions. For intracellular staining of

transcription factors, cells were fixed and permeabilized with fixation/permeabilization kit (ThermoFisher, cat # 00-5523), then stained with following antibodies; BLIMP1–AlexaFluor647 (clone: 5E7, Biolegend, cat # 150004, 1:300), Pax5-PerCP-CY5.5 (clone: 1H9, Biolegend, cat #649709, 1:300), XBP1s-PE (clone: Q3-695, BD Biosciences, cat # 562642, 1:30), and IRF4-PE (clone: 3E4, ThermoFisher, cat # 12-9858-80, 1:300). The samples were analyzed on LSR II flow cytometer (BD Biosciences). Cell sorting was performed on FACSAria II flow cytometer (BD Biosciences).

**ELISA and ELISpot assays.** Total levels of immunoglobulins of different isotypes in the sera or culture supernatants were quantified using SBA Clonotyping system-horseradish peroxidase (HRP) kit (Southern Biotech, cat # 5300-05) according to the manufacturer's recommendation. NP-specific IgM, IgG1, IgG2a, IgG2b, and IgG3 in the serum was measured using plates (Maxisorp, ThermoFisher, cat # 439454) coated with $NP_{20}$-BSA (Bioresearch Technologies, cat # N-5050H) and Clonotyping system-HRP kit (Southern Biotech). Pooled sera from NP-KLH-immunized WT mice, 14 days post immunization were used as standard control. ELISpot assays were performed for the detection of NP-specific ASCs. Multiscreen membrane filtration 96-well plates (EMD Millipore, cat # MAIPSWU10) were activated by 35% ethanol and coated with 25 μg/ml NP-BSA. Cells from spleen and bone marrow were seeded into wells with and incubated overnight at 37 °C. Wells were washed and developed using goat anti-mouse IgG1-HRP or goat anti-mouse Igκ-HRP (Southern Biotech, cat # 5300-05, 1:1000) and 3,3′-diaminobenzidine (Vectorlabs, cat # SK-4100). The spots were counted manually and photographed using ImmunoSpot analyzer (Cellular Technology Limited).

**Quantitative real-time PCR.** Total RNA was isolated from cells using RNeasy plus micro kit (Qiagen, cat # 74034) according to the manufacturer's instructions. cDNA was synthesized from RNA using SuperScript III first-strand kit (ThermoFisher, cat # 18080051) according to the manufacturer's instructions. qPCR was performed on cDNA using iTaq Universal SYBR Green kit (Bio-Rad, cat # 172–5121) and StepOnePlus Real-Time PCR system (ThermoFisher). Primer Sequences are listed in Supplementary Table 1.

**Cellular fractionation and western blot analysis.** Total cell lysates were prepared by lysing cells in Laemmli buffer containing β-mercaptoethanol. In some experiments, cells were fractionated into ER and cytosolic fractions. To prepare ER and cytosolic fractions, $80 \times 10^6$ cells were lysed and sonicated in MTE buffer (270 mM D-mannitol, 10 mM Tris pH 7.4, and 0.1 mM EDTA). Cellular debris and mitochondria were removed by sequential centrifugation at 700 and $15,000 \times g$. The clear supernatant was loaded on the top of the discontinuous sucrose gradient (1.3, 1.5, and 2 M) and ultra-centrifuged at $152,000 \times g$ for 70 min in an SW41 rotor (Beckman Coulter). Top layer was collected as cytosol. ER fraction was collected as band at the interphase of 1.3 M sucrose layer. In the next step, ER fraction was ultra-centrifuged with additional MTE buffer at $126,000 \times g$ for 45 min in SW60 Ti rotor and pellet was collected as purified ER. Samples were electrophoresed into SDS-polyacrylamide gel electrophoresis gel and transferred to 0.45 μM nitrocellulose membrane (GE Healthcare, cat # 10600002). Membrane were blotted with primary antibody and HRP-conjugated secondary antibody. The following antibodies were used: Ufbp1 (rabbit polyclonal, Proteintech, cat # 21445-1-AP, 1:2000), β-actin (Clone: AC-15, Sigma-Aldrich, cat # A5441, 1:30,000), PDI (Clone: 34/PDI, BD Biosciences, cat # 610946, 1:1000), IRE1α (Clone: 14C10, Cell Signaling, cat # 3294P, 1:1000), XBP1s (rabbit polyclonal, Biolegend, cat # 619502, 1:500), p-PERK (Clone: T980, Cell Signaling, cat # 3179, 1:1000), PERK (Clone: C33E10, Cell Signaling, cat # 3192, 1:1000), p-eIF2α (Clone: D9G8, Cell Signaling, cat # 3398, 1:1000), eIF2α (Clone: D7D3, Cell Signaling, cat # 5324, 1:1000), ATF4 (Clone: D4B8, Cell Signaling, cat # 11815, 1:1000), and Ufm1 (Abcam, cat # ab109305, 1:2000). Antibodies against Uba5 (1:500), Ufc1 (1:200), and Ufl1 (1:500) were obtained from Dr. Honglin Li laboratory (Augusta University) and published. All HRP-conjugated secondary antibodies were obtained from Jackson ImmunoResearch (goat anti-rabbit IgG-HRP, cat # 111-035-003, 1:5000; goat anti-mouse IgG-HRP, cat # 115-035-003, 1:5000; and goat anti-rat IgG-HRP, cat # 112-035-003, 1:5000). The uncropped images of important immunoblots are shown in Supplementary Figure 9.

**Immunofluorescence.** Cells were attached to poly-L-lysine (Sigma-Aldrich, cat # P1524)-coated glass slides. Cells were fixed with 4% paraformaldehyde (Sigma-Aldrich, cat # 158127) for 20 min at room temperature. Permeabilized cells were incubated with primary antibody at room temperature. Cells were stained with fluorescent-conjugated second antibody from Jackson ImmunoResearch (Goat anti-Rabbit IgG-AlexaFluor488, cat # 111-545-144, 1:1000; and Goat anti-Mouse IgG-AlexaFluor594, cat # 115-585-146, 1:1000) and 4′,6-diamidino-2-phenylindole (Sigma-Aldrich, cat # D9542). Images were acquired using Keyence digital microscope.

**Transmission electron microscopy.** Cells were fixed in 4% paraformaldehyde and 2% glutaraldehyde in 0.1 M sodium cacodylate. Postfixed cells were stained with osmium tetroxide, dehydrated, and embedded in Epon-Araldite resin. Ultrathin sections were sliced with Leica EM UC6 Ultramicrotome (Leica Microsystem) and collected on copper grids. The sections were sequentially stained with uranyl acetate and lead citrate and visualized by JEM1230 TEM (JEOL).

**Retroviral constructs, purification, and transduction.** cDNA from mouse XBP1u, XBP1s, and human Ufbp1, and its mutant Ufbp1K267R were cloned into retrovirus vector MSCV-PIG (Addgene, cat# 18751) and/or MSCV-Thy1.1 (Addgene, cat# 17442). Retroviruses were prepared using Plat-E packaging cell line (Cell Biolabs, Inc, cat # RV-101). Viral supernatants were concentrated by centrifugation at $10,000 \times g$ at 4 °C overnight. LPS-stimulated B cells were infected with retrovirus by centrifugation at $700 \times g$ for 30 min in the presence of 4 μg/ml polybrene. Two days after infection, cells were analyzed as indicated in figure legends.

**Statistical analysis.** Statistical significance was calculated using Student's t-test with two-tailed analysis.

**Reporting summary.** Further information on experimental design is available in the Nature Research Reporting Summary linked to this article.

## Data availability

All data are available from authors upon reasonable request.

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

## Acknowledgements

We thank S. Rath and D.H. Munn for critical reading of manuscript. This research was supported by National Institutes of Health grants R01 DK103185, R01 CA198103, and R01 DK113171 to R.J.K.; R01 DK113409 and NSFC 81570094 to H.L.; and American Association of Immunologists Careers in Immunology Fellowship and R01DK103576 to N.S. We also thank the Georgia Cancer Center Flow Cytometry Core and Electron Microscope and Histology Core of Augusta University for technical help.

## Author contributions

H.Z. helped conceive, design, perform, and interpret experiments and wrote the manuscript. B.B. provided help to H.Z. S.S. performed experiments. Y.C. and S.L. provided

technical help. S.K.K., N.P., N.M.S. and A.S. provided expertise and feedback. R.J.K. provided $Ire1\alpha^{F/F}$ mice, advice, and edited the manuscript. H.L. provided $Ufbp1^{F/F}$ mice, reagents, and expertise. N.S. conceived, designed, and interpreted experiments, wrote the manuscript, and secured funding.

## Additional information

**Competing interests:** The authors declare no competing interests.

