## [Peer Review File · Nature Communications]

Reviewers' comments:

Reviewer #1 (Plasma cell, Blimp1, B cell biology)(Remarks to the Author):

The study of Zhu and colleagues addresses the role of Ufbp1 in the UPR response in plasma cells (PCs). They propose that Ufbp1, both a target and itself involved in Ufmylation, is involved in plasma cell development, Ab secretion and ER expansion. Ufbp1 also suppresses the PERK/ATF4 arm of the UPR.

Overall I find this to be an interesting study that increases our knowledge of the regulation of the UPR in general and in PCs in particular. I do though have a serious reservation about the conclusion that Ufbp1 is involved in PC development (as stated in the title and major conclusion of the paper) and believe that this conclusion stems from limitations in the use of the marker CD138 to solely define PCs, leading to over-interpretation of the data. If the authors can convincingly sort out the issues raised below, I feel that would greatly improve the manuscript.

Major point

While the authors convincingly show that Ufbp1 is involved in Ab secretion and ER expansion in PCs, I am sceptical of the conclusion that PC development per se is impacted by Ufbp1-deficiency. Points 1-3 below lay out this concern in more detail.

1. Figure 1 A-C, 6A-D, S2B-C. The combination B220-CD138+ is not sufficient to identify PCs in vivo with any accuracy. Indeed it is known that many PCs are B220+ to varying degrees. This has been clearly shown using Blimp-1-GFP mice and more recently other markers, such as TACI (Pracht Eur J Immunol, 2017), Sca-1 (Wilmore Eur J Immunol, 2017) and CD98 (Shi Nat Imm 2015) have been combined with CD138. In addition in situations that don't require live cells to be analysed, intracellular staining for IRF4 (or Blimp-1) can also be used.

The authors need to use these additional markers to more convincingly identify and enumerate PCs. For example, it is known that the BM PC compartment is typically full in unimmunized (steady-state) mice, and immunization alters the make up of the compartment (after ~day 14) but not the total number of BM PCs. Thus the data in figure 1C isn't likely to be completely correct.

2. The analysis of LPS cultures in Figure 3 is also problematic for 2 reasons. LPS cultures are well known to produce Ab secreting cells that are both CD138+ and CD138- at a similar frequency (eg, Kallies JEM 2004). Both populations produce similar amount of Ab. It also appears that CD138 expression itself, and not PC number, are decreased upon Ufbp1 loss. This may be due to the increased sXbp1 as Taubenheim JI 2012 showed that the converse was true; CD138 expression of the surface of PCs increased in Xbp1/Cd19Cre mice. In terms of PC frequency, the authors own data in shows normal expression of Blimp-1 (Figure 3d, S3C), IRF4 (Figure S3D) and down-regulation Pax5 (Figure 3F) in the Ufbp1 KO. Thus I would suggest that by these criteria PC frequency is normal in these assays and it is CD138 expression is altered and I suggest that the authors either provide more convincing evidence that PC differentiation is impacted on the KO, or adjust their interpretation of this data.

3. Figure 6E-H. Similar issue. Authors should measure Blimp-1, Pax5 and or IRF4 to quantify PC numbers and ER expansion independent of CD138.

4. Figure 7E. Lacks a WT control. ER expansion is low in Ire1 KO, and so the "rescue" is hard to put in context without a WT comparison. Conceptually it is hard to imagine that Ufbp1 overexpression rescues Ire1 (and thus sXbp1) deficiency as the cells would also lack high expression of all the Xbp1 regulated genes in this pathway, as shown in figure 7.

5. What is the consequence of ectopic PERK overexpression in the LPS cultures? Does this copy Ufbp1 KO?

6. Figure S3H-I. I find the data in the Ufm1 blots unconvincing. The data suffers from the same criticism as point #2 in regards CD138 not being a good PC marker in LPS cultures. Moreover, as the virtually all the components of the Ufm1ylation pathway are strongly upregulated in PCs, it seems odd that there is no clear increase in Ufm1ylated protein in PCs, nor any clear pattern to the data. Short of developing a convincing way to isolate and compare activated B cells from PCs in this assay, I feel that this data should be removed from the paper.

Reviewer #2 (Post-translational modification, B cell biology)(Remarks to the Author):

Ufm1 conjugation system is an ubiquitin-like modification system. Ufbp1 is a putative Ufm1 target, but its physiological role remains largely unknown. This group previously showed that inducible deletion of Ufbp1 in adult mice caused impaired adult hematopoiesis and pancytopenia. This manuscript by Zhu et al further described the role of Ufbp1 in promoting plasma cell (PC) differentiation through suppressing the activation of PERK of UPR pathway. They showed that deletion of Ufbp1 in B cell lineage impairs the production of Ig, which is linked with the function of Ufbp1 in the expansion of endoplasmic reticulum (ER) in antibody secreting cells (ASCs). Mechanistically, the authors showed that Ufbp1 regulates IRE1 α and PERK pathways in UPR and that Ufm1 machinery is downstream to IRE1/XBP-1 pathway. Overall, UFBP1 deficiency led to elevated ER stress and activation of UPR has been demonstrated, somehow hampering the novelty of this manuscript. However, this study presents an impressive amount of data resulting from genetic disruption of the UPR pathway and Ufbp1 expression in B cells that appear to support the significance of Ufbp1 in ASC production. In order to ensure the findings and the involvement of Ufm1 conjugation of Ufbp1 is important in ASC production, and their other findings that may not be entirely consistent with previous literatures related to UPR in ASC generation, several points should be addressed:

Major:

1. The regulatory relationship between IRE1 α /XBP-1 and Ufbp1 described in this manuscript is rather confusing and should be further clarified. It is not convincing that IRE1 α /XBP-1 act upstream of Ufbp1, given the following reasons: 1) XBP1^{f/f}/XCD19Cre mice had markedly impaired Ig responses after NP-KLH immunization, but showed comparable numbers of B220⁺CD138⁺ splenocytes to the control littermates (J Exp Med. 2009 Sep 28; 206(10):2151-9). However, the observations made here demonstrated that the generation of B220⁺CD138⁺ plasma cells (PCs) in bone marrow and spleens following immunization and in the steady-state (Figure 2, Supplementary Figure 2 and others) is largely reduced in Ufbp1^{f/f}/XCD19Cre mice. 2) LPS stimulated B cells from IRE1 α ^{f/f}/XCD19Cre mice showed no expression of XBP-1, but the effect of lack of IRE1 α /XBP1 on the expression of various genes in the Ufm1 conjugation system, including Ufbp1, is only moderately affected (Figure 7 and Supplementary Figure 7). 3) However, in Figure 5c and d, the authors showed that the expression of IRE1 α and XBP-1 are both increased in stimulated B cells from Ufbp1^{f/f}/XCD19Cre mice. Whether there is a feedback loop is not clear and should be addressed.
2. It has been demonstrated that Perk^{-/-} B cells are competent for induction of Ig synthesis and antibody secretion (Mol Immunol. 2008 Feb; 45(4): 1035-43). The authors proposed that Ufbp1 silenced the PERK-dependent branch of UPR in ASC formation because Ufbp1 deficient stimulated B cells showed marginally increased phosphorylated levels of PERK (Figures 5f-h). The author further showed that Perk^{f/f}/Ufbp1^{f/f}/XCD19Cre B cells had normal frequency of B220⁺CD138⁺ PCs based on the expression of CD138, but Ig production remained defective (Figure 6). Those findings are rather

descriptive, lacking mechanistic insights. It is still unclear how Ufbp1 negatively regulates PERK and why PERK only affects CD138 expression.

3. The authors would like to link the role of Ufbp1 and Ufm1 conjugation system in ASC formation. Their data may only present the importance of Ufbp1 in ASC formation. The authors did not demonstrate the presence of Ufm1ylated Ufbp1 proteins and the abundance of Ufm1ylated Ufbp1 vs. un-Ufm1ylated Ufbp1 in the course of PC differentiation. Whether other Ufm1ylated proteins, such as ASC1, are also present? Any band in Supplementary Figure S3g represents Ufbp1? Further, the authors should try to evaluate the effect of K276R Ufbp1 on PC differentiation using B cells or mice with or without Ufbp1.

Minor:

Figure 1

Statistics analysis was not done in 1d, although the authors claimed that Ufbp1f/fXCD19Cre mice generate significantly reduced amounts of NP-specific IgG3.

Figure 2

1. Y axis in 2d is missing.

2. Ufbp1f/fXCD19Cre mice had more NP-specific GC B cells. The authors suggest a developmental block at GC in Ufbp1 deficient mice. This finding is difficult to reconcile the role of Ufbp1 in UPR and PC generation. The authors should also examine if Ufbp1 has additional function, not ER expansion, in GC B cells.

Figure 3 and Figure S3

1. The authors demonstrated the expression of various transcription factors, including Blimp-1, Pax5 and IRF4 by flow cytometric analysis. The appropriate controls, such as time course and isotype control antibody etc, should be included to demonstrate the specificity of staining.

2. In 3j, the expression of exogenous Ufbp1 should be included. Ig production should be analyzed.

Figure 4

1. 4e, the disruption of ER network should be indicated.

Figure 5 and Figure S5

1. Appropriate control should be included in 5a and b to demonstrate the expression of XBP-1 by using flow cytometric analysis.

2. The expression of XBP-1s and XBP-1u should be indicated/distinguished in 5c-e.

Figure 6

1. 6a, it is rather confusing why Perkf/fUfbp1f/fXCD19Cre B cells show normal CD138 frequency. The authors should further characterize mechanistically into this. Whether this is due to alteration of cell proliferation or survival?

2. 6i and j, ELISPOT data should be also included.

Figure 7

1. The expression of XBP-1u and XBP-1s should be provided in 7b.

2. The expression of exogenous XBP-1u and XBP-1s in 7d should be shown. To ensure that up-regulation of Ufm1 system and Ufbp1 is indeed downstream of Ire1 α /XBP-1, Ufm1 immunoblotting using total lysates from un-stimulated and stimulated Ire1 α f/fXCD19Cre and control B cells should be compared.

3. The expression of exogenous Ufbp1 and K267R Ufbp1 should be shown in 7e. The authors failed to show the frequency of CD138+ and Ig production in 7e.

We would like to thank the Reviewers for their valuable comments and suggestions. Please find below point-to-point responses to their critiques.

Reviewer #1 (Plasma cell, Blimp1, B cell biology)(Remarks to the Author):

The study of Zhu and colleagues addresses the role of Ufbp1 in the UPR response in plasma cells (PCs). They propose that Ufbp1, both a target and itself involved in Ufm1ylation, is involved in plasma cell development, Ab secretion and ER expansion. Ufbp1 also suppresses the PERK/ATF4 arm of the UPR.

Overall I find this to be an interesting study that increases our knowledge of the regulation of the UPR in general and in PCs in particular. I do though have a serious reservation about the conclusion that Ufbp1 is involved in PC development (as stated in the title and major conclusion of the paper) and believe that this conclusion stems from limitations in the use of the marker CD138 to solely define PCs, leading to over-interpretation of the data. If the authors can convincingly sort out the issues raised below, I feel that would greatly improve the manuscript.

Response: Using BLIMP1⁺CD138^{high} and IRF4⁺Pax5⁻ as markers for plasma cells, revised manuscript conclusively demonstrate that Ufbp1 regulates plasma cell development.

Major points

While the authors convincingly show that Ufbp1 is involved in Ab secretion and ER expansion in PCs, I am sceptical of the conclusion that PC development per se is impacted by Ufbp1-deficiency. Points 1-3 below lay out this concern in more detail.

Point 1. Figs. 2 A-C, 6A-D, S2B-C. The combination B220-CD138⁺ is not sufficient to identify PCs in vivo with any accuracy. Indeed it is known that many PCs are B220⁺ to varying degrees. This has been clearly shown using Blimp-1-GFP mice and more recently other markers, such as TACI (Pracht Eur J Immunol, 2017), Sca-1 (Wilmore Eur J Immunol, 2017) and CD98 (Shi Nat Imm 2015) have been combined with CD138. In addition in situations that don't require live cells to be analysed, intracellular staining for IRF4 (or Blimp-1) can also be used. The authors need to use these additional markers to more convincingly identify and enumerate PCs. For example, it is known that the BM PC compartment is typically full in unimmunized (steady-state) mice, and immunization alters the make up of the compartment (after ~day 14) but not the total number of BM PCs. Thus the data in figure 1C isn't likely to be completely correct.

Response: New Figs. 2a-d, 3a-d, 6a-f and Supplementary Figs. 2b-e and 6d show enumeration of plasma cells by BLIMP1⁺CD138^{high} and/or IRF4⁺Pax5⁻ staining. These data conclusively show a significant reduction of plasma cells in mice lacking Ufbp1 in B cells. Moreover, the reduction in plasma cells number in mice lacking Ufbp1 in B cells is rescued by the deletion of PERK in B cells (Figs 6a-f, 6d). The revised manuscript has referenced articles, which have combined CD138 with TACI, Sca-1 or CD98 to identify plasma cells. For experiments involving ER staining, which requires live cells, TACI⁺CD138^{high} cells were identified as plasma cells and plasmablasts (Figs. 6g-h, Supplementary figs. 4e, f, 6b, e). Using IRF4⁺Pax5⁻ cells as an identification marker for plasma cells, the new Figs 2c-d show that the bone marrow plasma cell compartment is comparable in immunized and unimmunized WT mice.

Point 2. The analysis of LPS cultures in Figure 3 is also problematic for 2 reasons. LPS cultures are well known to produce Ab secreting cells that are both CD138⁺ and CD138⁻ at a similar frequency (eg, Kallies JEM 2004). Both populations produce similar amount of Ab. It also appears that CD138 expression itself, and not PC number, are decreased upon Ufbp1 loss. This may be due to the increased sXbp1 as Taubenheim JI 2012 showed that the converse was true; CD138 expression of the surface of PCs increased in Xbp1/Cd19Cre mice. In terms of PC frequency, the authors own data in shows normal expression of Blimp-1 (Figure 3d, S3C), IRF4 (Figure S3D) and down-regulation Pax5 (Figure 3F) in the Ufbp1 KO. Thus I would suggest that by these criteria PC frequency is normal in these assays and it is CD138 expression is altered and I suggest that the authors either provide more convincing evidence that PC differentiation is impacted on the KO, or adjust their interpretation of this data.

Response: Previous data was obtained using a high dose of LPS (25 µg/ml). The revised Fig 3 using a combination of low dose of LPS (1µg/ml) + IL-4 + IL-5 demonstrates that 1) the development of BLIMP1⁺CD138^{high} and IRF4⁺Pax5⁻ cells are significantly decreased in the cultures started with Ufbp1-deficient B cells than Ufbp1-sufficient B cells, and 2) frequency of BLIMP1⁺ and IRF4⁺ cells are also decreased in the cultures of Ufbp1-deficient B cells than Ufbp1-sufficient B cells (Supplementary Figs 3a-b). Most importantly, similar data was obtained *in vivo*, when frequency of BLIMP1⁺CD138^{high} cells (plasma cells) among NP-specific cells after immunization with NP-KLH was evaluated (Figs 2i-j). These finding conclusively demonstrate that plasma cell development is defective in the absence Ufbp1.

Point 3. Figure 6E-H. Similar issue. Authors should measure Blimp-1, Pax5 and or IRF4 to quantify PC numbers and ER expansion independent of CD138.

Response: Using expression of BLIMP, IRF4 and Pax5, new Fig 6a-f and Supplementary Fig 6d show that deletion of PERK rescued the defective development of Ufbp1-deficient B cells into plasmablasts. Moreover, for ER staining, which requires analysis of live cells, TACI⁺CD138^{high} cells were identified as plasma cells and plasmablasts (Fig 6g-h, Supplementary Figure 4e, f, 6b, e)

Point 4. Figure 7E. Lacks a WT control. ER expansion is low in Ire1 KO, and so the “rescue” is hard to put in context without a WT comparison. Conceptually it is hard to imagine that Ufbp1 overexpression rescues Ire1 (and thus sXbp1) deficiency as the cells would also lack high expression of all the Xbp1 regulated genes in this pathway, as shown in figure 7.

Response: New Fig. 7e shows quantification of ER mass in IRE1α-sufficient and IRE1α-deficient plasmablasts and IRE1α-deficient plasmablasts overexpressing Ufbp1. When compared with ER mass in WT cells, Ufbp1 only partially rescued the ER mass in IRE1α-deficient plasmablasts. In line with the reviewer’s view we have added the following sentence in the results section on page 17: Alternatively, the partial rescue of ER mass in Ufbp1 expressing IRE1α-deficient plasmablasts may be due to lower expression of other XBP1 targets (such as PDI, EDEM1, ERDj4, Sec61a etc) involved in ER expansion.

Point 5. What is the consequence of ectopic PERK overexpression in the LPS cultures? Does

this copy Ufbp1 KO?

Response: Accumulation of ATF4 is the first stable product of the activation of the PERK pathway. A higher level of ATF4 was present in LPS-activated Ufbp1-deficient B cells than WT B cells. In contrast, the ATF4 level between WT, PERK-deficient B cells and B cells lacking both PERK and Ufbp1 was comparable. These findings suggest that Ufbp1 completely suppresses PERK activation during the development of plasma cells. These data suggest that activation of PERK rather than its expression level suppress development of plasma cells. Since there is no activation of PERK in WT mice and overexpression of PERK does not translate into active PERK, we do not think that simple overexpression of PERK will affect the development of plasma cells.

Point 6. Figure S3H-I. I find the data in the Ufm1 blots unconvincing. The data suffers from the same criticism as point #2 in regards CD138 not being a good PC marker in LPS cultures. Moreover, as the virtually all the components of the Ufmylation pathway are strongly upregulated in PCs, it seems odd that there is no clear increase in Ifmylated protein in PCs, nor any clear pattern to the data. Short of developing a convincing way to isolate and compare activated B cells from PCs in this assay, I feel that this data should be removed from the paper.

Response: We agree with the reviewer and the new manuscript does not contain this data.

Reviewer #2 (Post-translational modification, B cell biology) (Remarks to the Author):

Ufm1 conjugation system is an ubiquitin-like modification system. Ufbp1 is a putative Ufm1 target, but its physiological role remains largely unknown. This group previously showed that inducible deletion of Ufbp1 in adult mice caused impaired adult hematopoiesis and pancytopenia. This manuscript by Zhu et al further described the role of Ufbp1 in promoting plasma cell (PC) differentiation through suppressing the activation of PERK of UPR pathway. They showed that deletion of Ufbp1 in B cell lineage impairs the production of Ig, which is linked with the function of Ufbp1 in the expansion of endoplasmic reticulum (ER) in antibody secreting cells (ASCs). Mechanistically, the authors showed that Ufbp1 regulates IRE1 α and PERK pathways in UPR and that Ufm1 machinery is downstream to IRE1 α /XBP-1 pathway. Overall, UFBP1 deficiency led to elevated ER stress and activation of UPR has been demonstrated, somehow hampering the novelty of this manuscript. However, this study presents an impressive amount of data resulting from genetic disruption of the UPR pathway and Ufbp1 expression in B cells that appear to support the significance of Ufbp1 in ASC production. In order to ensure the findings and the involvement of Ufm1 conjugation of Ufbp1 is important in ASC production, and their other findings that may not be entirely consistent with previous literatures related to UPR in ASC generation, several points should be addressed:

Response: This manuscript contains several novel findings; 1) Ufbp1 positively regulates plasma cell development, 2) Ufbp1 regulates plasma cell development by suppressing PERK activation, 3) Ufbp1 positively regulates ER expansion in plasma cells, 4) Ufbp1 expression in

plasma cells is downstream of IRE1 α /XBP1 and 5) Ufbp1 links IRE1 α /XBP1 to ER expansion in plasma cells.

We do not find any data presented in the manuscript is inconsistent with previous literature related to the role of UPR in the generation of plasma cells. The failure of previous studies to find the suppressive role of PERK in development of plasma cells is well explained by the complete suppression of PERK branch of UPR (in our studies as well those published by others such as Ma et al., *Cell stress & chaperones* 2009 15:281, Goldfinger et al., *European journal of immunology* 2011, 41:491) by Ufbp1 (Fig. 6a). Accumulation of ATF4 is the first stable product of the activation of PERK pathway. A higher level of ATF4 was present in LPS-activated Ufbp1-deficient B cells than WT B cells. In contrast, ATF4 level between WT B cells and B cells lacking either PERK or both PERK and Ufbp1 B cells was comparable. These findings suggest that Ufbp1 completely suppresses PERK activation during the development of plasma cells. Therefore, single deficiency of PERK does not affect generation of plasma cells.

The findings that there is no defect in the development of plasma cells in *XBP1^{F/F}XCD19^{Cre}* mice, whereas there is a blockade of plasma cell development in *Ufbp1^{F/F}CD19^{cre}* mice demonstrates that Ire1 α /XBP1-independent expression of Ufbp1 (Supplementary Fig 7b shows ~48% of Ufbp1 protein expression is Ire1 α /XBP1-independent, whereas ~52% is Ire1 α /XBP1-dependent) is enough to drive plasma cell development in *XBP1^{F/F}XCD19^{Cre}* mice.

Major Points.

Point 1. The regulatory relationship between Ire1 α /XBP-1 and Ufbp1 described in this manuscript is rather confusing and should be further clarified. It is not convincing that Ire1 α /XBP-1 act upstream of Ufbp1, given the following reasons: **1)** *XBP1f/fXCD19Cre* mice had markedly impaired Ig responses after NP-KLH immunization, but showed comparable numbers of B220INTCD138+ splenocytes to the control littermates (J Exp Med. 2009 Sep 28;206(10):2151-9). However, the observations made here demonstrated that the generation of B220loCD138+ plasma cells (PCs) in bone marrow and spleens following immunization and in the steady-state (Figure 2, Supplementary Figure 2 and others) is largely reduced in *Ufbp1f/fXCD19Cre* mice. **2)** LPS stimulated B cells from *Ire1 α f/f XCD19Cre* mice showed no expression of XBP-1, but the effect of lack of Ire1 α /XBP1 on the expression of various genes in the Ufm1 conjugation system, including Ufbp1, is only moderately affected (Figure 7 and Supplementary Figure 7). **3)**

However, in Figure 5c and d, the authors showed that the expression of IRE1 α and XBP-1 are both increased in stimulated B cells from *Ufbp1f/fXCD19Cre* mice. Whether there is a feedback loop is not clear and should be addressed.

Response: *During mechanistic study when a biological function/process is considered as endpoint such as ER expansion in plasma cells, defining upstream and downstream events are important. A series of data presented in this manuscript conclusively show that if ER expansion in plasma cells is taken as endpoint of IRE1/XBP1 and Ufbp1 pathway, IRE1 α /XBP1 is upstream of Ufbp1. These data are briefly described as following.*

- a) Expression of the mRNAs of Ufbp1 and other genes involved in ufmylation is significantly (not totally) reduced in plasma cells and LPS-activated B cells from *Ire1 α ^{F/F}CD19^{cre}* mice than WT mice (Figs. 7a and c).

- b) Expression of protein levels of Ufbp1 and other molecules involved in ufmylation pathway is significantly (not totally) reduced in LPS-activated B cells from *Ire1α^{F/F}CD19^{cre}* mice than WT mice (Fig.7b and supplementary Fig. 7a).
- c) Expression of spliced XBP1 (product of IRE1α activation) rather than unspliced XBP1 significantly rescued the reduced expression of mRNAs of Ufbp1 and other genes involved in ufmylation pathway in plasmablasts from *Ire1α^{F/F}CD19^{cre}* mice to the WT level (Fig 7d).
- d) Immunoblotting using anti-Ufm1 antibody demonstrates a reduction in activity of ufmylation pathway in LPS activated B cells from *Ire1α^{F/F}CD19^{cre}* mice than WT mice (Supplementary Fig 7b).
- e) Like IRE1α/XBP1-deficient plasma cells or plasmablasts, Ufbp1-deficient plasma cells or plasmablasts show defective ER expansion (Fig 4c-f, 7e).
- f) Defective ER expansion in IRE1α-deficient plasmablasts is significantly rescued by overexpression of Ufbp1 (Fig 7e).
- g) Ufbp1-K267R, a Ufbp1-mutant which fails to undergo ufmylation, does not expand ER in IRE1α-deficient plasmablasts

Moreover, our conclusion that a significant fraction of Ufbp1 and other genes involved in ufmylation pathway are downstream to IRE1α/XBP1-pathway is also supported by a recent study (Tellier et al Nature Immunology, 2016) from Dr. Nutt group. RNA-seq data in Supplementary table 3 of this study shows that all the genes involved in ufmylation pathway; Uba5, Ufc1, Ufl1, Ufbp1(DDRGK1) and Ufm1 were significantly reduced in plasma cells lacking XBP1 than WT plasma cells.

Reviewer's confusion and clarifications

The reviewer assumes that our manuscript shows that the expression of Ufbp1 is 100% dependent on IRE1α/XBP1. This assumption has led to confusion in his/her interpretation of data shown in manuscript. The manuscript does not state/show anywhere that expression of Ufbp1 (and other genes involved in ufmylation) is 100% dependent on IRE1α/XBP1. The manuscript shows that in naïve B cells, expression of Ufbp1 (and other genes involved in ufmylation) is independent of IRE1α/XBP1 (Fig 7a). In plasma cells and LPS-activated B cells, the expression of a significant fraction (not total) of Ufbp1 (and other genes involved in ufmylation) is downstream to IRE1α/XBP1 (Fig 7a, b, c, d and, Supplementary Figs 7a and b). For example, at the protein level, 48%, 34%, 52%, 43% and 60% expression of Ufbp1, Uba5, Ufc1, Ufl1 and Ufm1 respectively are IRE1α/XBP1-independent, and whereas the rest of their expressions are IRE1α/XBP1-dependent in LPS-activated B cells (Supplementary Figs 7a). Thus, there are IRE1α/XBP1-independent and -dependent pools of Ufbp1 (and other genes involved in ufmylation) in B cells and plasma cells. (The underlined text in this paragraph has been added to the text on page 16 and 17).

Reviewer has given three reasons (as mentioned in point 1) that doubts whether Ufbp1 is downstream of IRE1α/XBP1. Point by point responses to these reasons are follows

- 1) Findings that there is no defect in the development of plasma cells in *XBP1^{F/F}XCD19^{Cre}* mice, whereas there is a blockade of plasma cell development in *Ufbp1^{F/F}CD19^{cre}* mice demonstrates a role of IRE1α/XBP1-independent expression of Ufbp1 in plasma cell

development. However, manuscript shows that 100% expression of Ufbp1 is not independent of IRE1 α /XBP1. Expression of ~ 58% mRNA and 52% protein of Ufbp1 in plasma cells and LPS-activated B cells respectively is IRE1 α /XBP1-dependent (Fig 7a, b, Supplementary Fig 7a). Supplementary Fig 7b shows reduced ufmylation activity IRE1 α -deficient B cells. Expression of spliced XBP1 rather than unspliced XBP1 significantly rescued the reduced expression of mRNAs of Ufbp1 and other genes involved in ufmylation pathway in plasmablasts from *Ire1 α ^{F/F}CD19^{cre}* mice to the WT level (Fig 7d). Like IRE1 α /XBP1-deficient plasma cells or plasmablasts, Ufbp1-deficient plasma cells or plasmablasts show defective ER expansion (Fig 4c-f, 7e). Moreover, defective ER expansion in IRE1 α -deficient plasmablasts is significantly rescued by overexpression of Ufbp1 (Fig 7e). Ufbp1-K267R, a Ufbp1-mutant which fails to undergo ufmylation, does not expand ER in IRE1 α -deficient plasmablasts (Fig 7e). Taken together, these data unequivocally demonstrate that expression of a significant fraction of Ufbp1 and other genes involved in ufmylation pathway downstream of IRE1 α /XBP1 play an essential role in expansion of ER.

2) LPS-stimulated B cells and plasma cells from *Ire1 α ^{F/F}CD19^{cre}* mice show a moderate downmodulation of Ufbp1 and other genes involved in ufmylation pathway (Fig 7a, b, c, and Supplementary Fig 7a). However, this downmodulation is **significant** (please see the statistics in Figure 7) **and was present at both mRNA, protein level**. This downmodulation is **significantly rescued** (please see the statistics in Figure 7) by expression of spliced XBP1 rather than unspliced XBP1 to the level seen in WT cells (Fig 7d). The activity of ufmylation pathways is reduced in LPS-activated IRE1 α -deficient B cells than WT B cells (Supplementary Fig 7b). These data perfectly conclude that expression of a significant fraction of Ufbp1, Uba5, Ufc1, Ufl1 and Ufm1 are downstream to IRE1 α /XBP1.

3) In cells lacking XBP1 or IRE1 α (IRE1 α knock out construct only lacks C terminal endonuclease domain and thus expresses a truncated IRE1 α protein) expression of IRE1 α is increased (Benhamron et al, *European journal of immunology* 2014, 44:867, Lee et al, *Science* 2008, 320:1492). However, none of these or other articles (despite being ~ hundreds of articles regarding relation between IRE1 α and XBP1) interpret that XBP1 is upstream of IRE1 α (due to splicing of XBP1 mRNA by IRE1 α , they always interpret that XBP1 is downstream of IRE1 α), because the increased expression of IRE1 α in the absence of XBP1 does not lead to the final outcome of IRE1 α activation; the expansion of ER. Similarly, the increased expression of IRE1 α in the absence of Ufbp1 (Figs 5c, d) does not lead to the final outcome of IRE1 α activation; the expansion of ER in Ufbp1-deficient plasm cells (Fig 4) and therefore it is difficult to interpret that Ufbp1 is upstream of IRE1 α .

These findings may suggest existence of a negative feedback loop where a compensatory mechanism under the conditions of defective ER expansion such as deficiency of either IRE1 α , XBP1 or Ufbp1 enhances IRE1 α expression. In any feedback loop where there is no defined start or end points, it would be difficult to ascertain which point/step is upstream or downstream. However, during mechanistic study when a biological function/process is considered as endpoint such as ER expansion in plasma cells, defining upstream and downstream events are important. A series of data presented in the manuscript conclusively show that if ER expansion in plasma cells is as an endpoint of IRE1/XBP1 and Ufbp1 pathway, then IRE1 α /XBP1 is upstream of Ufbp1.

Point 2. It has been demonstrated that Perk^{-/-} B cells are competent for induction of Ig synthesis

and antibody secretion (Mol Immunol. 2008 Feb;45(4):1035-43). The authors proposed that Ufbp1 silenced the PERK-dependent branch of UPR in ASC formation because Ufbp1 deficient stimulated B cells showed marginally increased phosphorylated levels of PERK (Figures 5f-h). The author further showed that *Perk^{f/f}Ufbp1^{f/f}XCD19^{Cre}* B cells had normal frequency of B220^{lo}CD138⁺ PCs based on the expression of CD138, but Ig production remained defective (Figure 6). Those findings are rather descriptive, lacking mechanistic insights. It is still unclear how Ufbp1 negatively regulates PERK and why PERK only affects CD138 expression.

Response: Fig 5 f-h shows that there was ~ 5-, 3- and 10-fold upregulation of phosphorylation of PERK (p-PERK), phosphorylation of eIF2 α (p-eIF2 α) and ATF4 respectively in Ufbp1-deficient B cells on day 1. To call these numbers **moderate** may be a subjective decision. The real question is whether this activation has any functional relevance? Using B cell lacking PERK and Ufbp1, Fig 6a shows that upregulation of ATF4 in Ufbp1-deficient activated B cells is indeed dependent on PERK. Whether, activation of PERK in Ufbp1-deficient B cells is just a phenomenon, or does it has a functional consequence? The answer is yes; because in mice lacking both PERK and Ufbp1 in B cells (*Perk^{F/F}Ufbp1^{F/F}XCD19^{Cre}* mice), frequencies of plasma cells and development of plasmablasts are comparable to WT mice (Figs. 6a-f and Supplementary Fig. 6c). Therefore, using genetic experiments and in vivo models, current manuscript mechanistically links suppression of PERK activation by Ufbp1 to the development of plasma cells.

However, expansion of ER in plasma cells lacking both Ufbp1 and PERK is defective (Fig 6g-i and Supplementary Fig. 6b and e). A well-developed ER network is required for Ig production as demonstrated by previous studies (*Van Anken et al., Immunity, 2003, 18:243, Todd et al., J Exp Med. 2009, 206:2151, Shaffer et al, Immunity. 2004, 21:81*). Therefore, lack of a well-developed ER (Figs. 6g-i and Supplementary Fig 6b and e) is the underlying mechanism of defective Ig production by plasma cells lacking both Ufbp1 and PERK and decreased levels of immunoglobulins in the serum of *Perk^{F/F}Ufbp1^{F/F}XCD19^{Cre}* mice.

Understanding the mechanism underlying Ufbp1-mediated suppression of PERK is important. However, it is out of the scope of current manuscript.

CD138 is just a marker for enumeration of plasma cells. PERK does not affect CD138 expression. Fig. 6 shows that defective development of Ufbp1-deficient B cells into plasma cells is rescued by deletion of PERK in them. Therefore, mice lacking both Ufbp1 and PERK in B cells have significantly higher number of BLIMP⁺CD138^{high} cells (plasma cells) than mice lacking only Ufbp1 in B cells.

Point 3. The authors would like to link the role of Ufbp1 and Ufm1 conjugation system in ASC formation. Their data may only present the importance of Ufbp1 in ASC formation. The authors did not demonstrate the presence of Ufm1ylated Ufbp1 proteins and the abundance of Ufm1ylated Ufbp1 vs. un-Ufm1ylated Ufbp1 in the course of PC differentiation. Whether other Ufm1ylated proteins, such as ASC1, are also present? Any band in Supplementary Figure S3g represents Ufbp1? Further, the authors should try to evaluate the effect of K276R Ufbp1 on PC differentiation using B cells or mice with or without Ufbp1.

Response: LPS induces differentiation of B cells into plasma cells. During LPS-activation, Supplementary Fig 7b shows that several ufm1ylated bands of approximate molecular weight (such as 28, 41, 50 kDa) were reduced in Ufbp1-deficient B cells than WT counterparts. Thus,

ufmylation activity is induced during course of plasma cell development. Identity of these bands and their role (as well as role of ASC1) in plasma cells development is beyond the scope of current manuscript.

Fig 3e and f shows that the ability of Ufbp1K267R, a mutant of Ufbp1 in which lysine at position 267 has been changed to arginine and thus does not undergo ufmylation, to promote plasma cell development is comparable to WT Ufbp1 molecule. Since lysine 267 (K267) is the main lysine in Ufbp1 that undergoes ufmylation (Tatsumi et al., J. Biol Chem 2010, 285:5417), this data suggest that Ufbp1 promotes plasma cell differentiation independent of its ufmylation at lysine 267. However, we cannot rule out the role of potential ufmylation of other conserved lysines at positions 116, 121, 124, 128, 193, 224 and 227 of Ufbp1 in differentiation of plasma cells. Alternatively, it is also possible that Ufbp1 promotes plasma cell development independent of its ufmylation.

Supplementary Fig 3e shows the Ig production by sorted GFP⁺CD138⁺ cells expressing either WT Ufbp1 or Ufbp1-K267R. WT Ufbp1 significantly rescued the Ig production by Ufbp1-deficient plasma cells whereas Ufbp1-K267R failed to do so. Therefore, lysine 267 of Ufbp1 distinguishes between role of Ufbp1 in plasma cell development versus its role in immunoglobulin production. Fig 7e demonstrate that Ufbp1-K267R failed to induce expansion of ER, which is required for efficient production of Ig (*Van Anken et al., Immunity, 2003, 18:243, Todd et al., J Exp Med. 2009, 206:2151, Shaffer et al, Immunity. 2004, 21:81*). Therefore, inability of Ufbp1-K267 to expand ER is consistent with its failure to enhance production of IgM by Ufbp1-deficient plasmablasts in Supplementary Fig 3e.

Minor:

Figure 1

Statistics analysis was not done in 1d, although the authors claimed that Ufbp1f/fXCD19Cre mice generate significantly reduced amounts of NP-specific IgG3.

Response: We apologize for missing statistics. The revised manuscript contains the statistics for Fig 1d.

Figure 2

1. Y axis in 2d is missing.

Response: We apologize for missing label for Y-axis. The revised manuscript contains the label for Fig 2d.

2. Ufbp1f/fXCD19Cre mice had more NP-specific GC B cells. The authors suggest a developmental block at GC in Ufbp1 deficient mice. This finding is difficult to reconcile the role of Ufbp1 in UPR and PC generation. The authors should also examine if Ufbp1 has additional function, not ER expansion, in GC B cells.

Response: The current manuscript conclusively shows the role of Ufbp1 in plasma cell development and expansion of ER network in them. Addressing the relation between Ufbp1, germinal center reaction, UPR and PC generation is out of scope of current manuscript.

Figure 3 and Figure S3

1. The authors demonstrated the expression of various transcription factors, including Blimp-1, Pax5 and IRF4 by flow cytometric analysis. The appropriate controls, such as time course and isotype control antibody etc, should be included to demonstrate the specificity of staining.

Response: The revised Figure 3a and c shows staining for isotype control antibody and the specificity of staining. Optimum time for appearance of plasmablasts in these cultures is day 4 (Tsu et al., *Immunity* 2018, 48:1144). Therefore, development of plasmablasts was analyzed on day 4 in Fig 3.

2. In 3j, the expression of exogenous Ufbp1 should be included. Ig production should be analyzed.

Response: Supplementary Fig 3d shows the expression of exogenous Ufbp1 in retrovirally transduced cells. Fig 3e shows the Ig production by sorted GFP⁺CD138⁺ cells. WT Ufbp1 significantly rescued the Ig production by Ufbp1-deficient plasma cells whereas Ufbp1-K267R failed to do so. Therefore, lysine 267 of Ufbp1 distinguishes between role of Ufbp1 in plasma cell development versus its role in immunoglobulin production. Fig 7e demonstrate that Ufbp1-K267R failed to induce expansion of ER, which is required for efficient production of Ig (*Van Anken et al., Immunity, 2003, 18:243, Todd et al., J Exp Med. 2009, 206:2151, Shaffer et al, Immunity. 2004, 21:81*). Therefore, inability of Ufbp1-K267 to expand ER is consistent with its failure to enhance production of IgM by Ufbp1-deficient plasmablasts in Supplementary Fig 3e.

Figure 4

1. 4e, the disruption of ER network should be indicated.

Response: Fig 4e shows that overall there is less ER network in the **entire** cytoplasm of Ufbp1-deficient plasmablasts. Therefore, it is difficult to point out a specific area/location that shows disruption of ER network. If we have to point anywhere, it will be entire cytoplasm, which is a big area in the cell and therefore is hard to be indicated by a point.

Figure 5 and Figure S5

1. Appropriate control should be included in 5a and b to demonstrate the expression of XBP-1 by using flow cytometric analysis.

Response: Revised figure 5a and 5b show isotype control antibody staining and confirms higher expression of XBP1s in Ufbp1-deficient plasma cells

2. The expression of XBP-1s and XBP-1u should be indicated/distinguished in 5c-e.

Response: XBP1s is a ~ 54kDa protein. XBP1u is a ~ 32 kDa protein. The molecular weight of XBP1 band shown in Fig 5c-e is ~ 54 kDa. The antibody used for detection of XBP1 in Fig 5 c-e is from Biolegend (cat # poly 6195) and only recognizes XBP1s. Therefore, the Fig 5c-e only shows XBP1s. XBP1u has a very short half-life and gets rapidly degraded by proteasome (Navon et al., *FEBS Letter, 2010, 584:67*). In addition, no role of XBP1u protein in plasma cells or B cells has been described. Therefore, measuring XBP1u will not result in meaningful data.

Figure 6

1. 6a, it is rather confusing why *Perk^f/Ufbp1^f/fXCD19Cre* B cells show normal CD138 frequency. The authors should further characterize mechanistically into this. Whether this is due to alteration of cell proliferation or survival?

Response: CD138 was used as a marker for identification of plasma cells in Fig 6a. Using additional markers such as BLIMP1, the revised Fig 6a-f and Supplementary Figure 6d confirm that deletion of PERK rescues the defective plasma cell development in mice lacking *Ufbp1* in B cells. Fig 5f-h and Supplementary Fig. 6a shows the activation of PERK in *Ufbp1*-deficient B cells. Therefore, it is logical that deletion of PERK restored the defective development of *Ufbp1*-deficient B cells into plasma cells (Fig 6 a-f, and supplementary Fig 6d). Supplementary figure 6c shows that a similar number of live cells were recovered during culture of WT, PERK-deficient, *Ufbp1*-deficient, and PERK- and *Ufbp1*- double deficient B cells. In addition, Supplementary Fig. 5j demonstrate that apoptosis of LPS-activated WT and *Ufbp1*-deficient B cells are comparable. These data suggest that deletion of PERK does not alter the proliferation or survival of *Ufbp1*-deficient cells.

2. 6i and j, ELISPOT data should be also included.

Response: ELISpot for immunoglobulin secretion by plasmablasts is shown in fig 6l and supplementary fig. 6f

Figure 7

1. The expression of XBP-1u and XBP-1s should be provided in 7b.

Response: XBP1s is a ~ 54kDa protein. XBP1u is a ~ 32 kDa protein. The molecular weight of XBP1 band shown in Fig 7b is ~ 54 kDa. The antibody used for detection of XBP1 is from Biologend (cat # poly 6195) and only recognizes XBP1s. Therefore, the Fig 7b only shows XBP1s. XBP1u has a very short half-life and gets rapidly degraded by proteasome (Navon et al., FEBS Letter, 2010, 584:67). In addition, no role of XBP1u protein in plasma cells or B cells has been described. Therefore, measuring XBP1u will not result in a meaningful data.

2. The expression of exogenous XBP-1u and XBP-1s in 7d should be shown. To ensure that up-regulation of Ufm1 system and *Ufbp1* is indeed downstream of *Ire1 α /XBP-1*, Ufm1 immunoblotting using total lysates from un-stimulated and stimulated *Ire1 α f/fXCD19Cre* and control B cells should be compared.

Response: Expression of XBP1u and XBP1s in retrovirally transduced cells used in Fig 7d are shown in supplementary Fig 7c. The data shows no splicing of XBP1 in *IRE1 α* -deficient B cells. Using Ufm1 immunoblotting on total cell lysates, supplementary Fig 7b shows decreased ufmylation activity in LPS-activated *IRE1 α* -deficient B cells than WT counterparts. These data together with other data in Fig. 7 demonstrate that *Ufbp1* and Ufm1 system is downstream of *IRE1 α /XBP1*

3. The expression of exogenous Ufbp1 and K267R Ufbp1 should be shown in 7e. The authors failed to show the frequency of CD138+ and Ig production in 7e.

Response: Supplementary figure 3b shows the ability of respective retroviruses to express Ufbp1 or its mutant. Revised Fig 7e shows that Ufbp1 partially rescues the ER expansion and Ufbp1K267R failed to do so in IRE1 α -deficient plasmablasts. IRE1/XBP1 regulate Ig production at multiple levels, e.g. ER expansion, degradation of I μ mRNA by RIDD (Benhamron et al, *European journal of immunology* 2014, 44:867), transcriptional activation of Igs promoter (Tellier et al *Nature Immunology*, 2016). XBP1 deficiency does not affect development of plasmblast (CD138+ cells) (Taubenheim, et al., *Journal of immunology* 2012, **189**:3328 and Todd et al., *J Exp Med.* 2009, 206:2151). Therefore, measuring Ig production and frequency of CD138+ cells may not generate any meaningful results.

Reviewers' comments:

Reviewer #1 (Remarks to the Author):

I feel the authors have done a thorough job of responding to my comments/concerns and I feel that this interesting manuscript is now suitable to publish as is.

The new data using BLIMP1, Pax5 and Irf4 intracellular FACS is appreciated and particularly convincing.

sincerely

Stephen Nutt

Reviewer #2 (Remarks to the Author):

The authors have adequately addressed my previous comments/concerns by completing additional experiments, providing additional explanation in the text or in the point-by-point reply, and correcting the errors in the previous manuscript. This study is recommended for publication.